# Biophysical basis of filamentous phage tactoid-mediated antibiotic tolerance in *P. aeruginosa*

Jan Böhning [1], Miles Graham [1,2], Suzanne C. Letham [1,2], Luke K. Davis [3,4], Ulrike Schulze[5], Phillip J. Stansfeld [6], Robin A. Corey [7,8], Philip Pearce [3,4], Abul K. Tarafder [1] ✉ & Tanmay A. M. Bharat [1] ✉

Inoviruses are filamentous phages infecting numerous prokaryotic phyla. Inoviruses can self-assemble into mesoscale structures with liquid-crystalline order, termed tactoids, which protect bacterial cells in *Pseudomonas aeruginosa* biofilms from antibiotics. Here, we investigate the structural, biophysical, and protective properties of tactoids formed by the *P. aeruginosa* phage Pf4 and *Escherichia coli* phage fd. A cryo-EM structure of the capsid from fd revealed distinct biochemical properties compared to Pf4. Fd and Pf4 formed tactoids with different morphologies that arise from differing phage geometries and packing densities, which in turn gave rise to different tactoid emergent properties. Finally, we showed that tactoids formed by either phage protect rod-shaped bacteria from antibiotic treatment, and that direct association with a tactoid is required for protection, demonstrating the formation of a diffusion barrier by the tactoid. This study provides insights into how filamentous molecules protect bacteria from extraneous substances in biofilms and in host-associated infections.

Viruses that infect bacteria and archaea, known as phages, are among the most common biological entities on Earth[1,2]. Filamentous bacteriophages, belonging to the family *Inoviridae*, are one of the major categories of phages, being pervasive in prokaryotes across Earth's biomes[3]. Inoviruses consist either of a linear single-stranded[4] or circular single-stranded DNA genome[5] bound by a filamentous array of an α-helical capsid protein. These rod-shaped inoviruses are typically ~60–70 Å in diameter, but, depending on their genome size, their length can vary from one to a few microns, as it is proportional to the size of the encapsidated genome. Typically, inoviruses do not lyse cells[6] and either integrate into the host cell genome as prophages (seen for phages Pf4, CTXφ and MDAφ), or are expressed by the host cell

episomally, i.e., not integrated into the bacterial genome (for fd, M13 and Pf1 phages)[2].

The first inovirus identified, named fd, was reported in 1963 as a DNA-containing phage with an unusual filamentous morphology infecting *Escherichia coli*[6,7]. Fd has since become a model for filamentous phage structure, infection, and assembly[7,8], as well as having found use in numerous biotechnological applications, including the first described instance of phage display[9], and as a model for examining colloidal-rod phase transitions[10]. Despite its importance in multiple areas of research, past structural studies have reported markedly differing atomic structures of the fd phage capsid (protein pVIII in fd) using fibre diffraction[11,12], solid-state nuclear magnetic resonance

[1]Structural Studies Division, MRC Laboratory of Molecular Biology, Francis Crick Avenue, Cambridge CB2 0QH, UK. [2]Sir William Dunn School of Pathology, University of Oxford, Oxford OX1 3RE, UK. [3]Department of Mathematics, University College London, London WC1H 0AY, UK. [4]Institute for the Physics of Living Systems, University College London, London WC1E 6BT, UK. [5]Cell Biology Division, MRC Laboratory of Molecular Biology, Francis Crick Avenue, Cambridge CB2 0QH, UK. [6]School of Life Sciences & Department of Chemistry, University of Warwick, Coventry, UK. [7]Department of Biochemistry, University of Oxford, Oxford OX1 3QU, UK. [8]School of Physiology, Pharmacology and Neuroscience, University of Bristol, Bristol BS8 1TD, UK. ✉e-mail: atarafder@mrc-lmb.cam.ac.uk; tbharat@mrc-lmb.cam.ac.uk

(ssNMR)[11,13] and 8 Å-resolution electron cryomicroscopy (cryo-EM)[14], meaning that the structure of the fd virion remains controversial. Thus far, only two high-resolution cryo-EM structures of filamentous phage capsids have been solved; the first from Ike, which like fd, is a member of the class I type of inoviral bacteriophages with pentameric (C5) capsid symmetry[5]; and second from Pf4, a class II inoviral bacteriophage with C1 symmetry, which is expressed as a prophage in *P. aeruginosa* biofilms[4].

A unique property of inoviral phages, including fd and Pf4, is that they can spontaneously and reversibly self-assemble to form ordered but dynamic structures termed tactoids under crowding conditions[15–17]. Within these tactoids, laterally associated phages are orientationally aligned, but not regularly ordered as in a crystal, resulting in local liquid-crystalline order[16]. Phages within tactoids are mobile, as shown by fluorescence recovery after photobleaching (FRAP) experiments on Pf4 tactoids[4]. Assembly of phages into tactoids is driven by a depletion interaction, an effective attractive force between rigid constituents arising from the preferential exclusion of solvent elements (depletants) from the vicinity of the larger phage constituents[17,18]. In the presence of high-molecular weight crowding polymers as a depletant and counterions to compensate surface charges, phages spontaneously assemble into liquid-crystalline tactoids. As expected from a depletion interaction, the size of the crowding polymer positively correlates with the extent of tactoid formation[15]. This process is thermodynamically favourable as phage alignment allows for higher degrees of freedom of the crowding polymer, increasing the total entropy[17,19].

Due to their monodisperse nature, filamentous phages have been used to study phase behaviour of rod-like molecules, with fd having been employed for this purpose for more than half a century[20–26]. More recently, studies have shown that filamentous phages are highly expressed in bacterial biofilms formed by the human pathogen *P. aeruginosa*[27]. For example, the phage Pf4 is integrated into the *P. aeruginosa* genome, where it is highly upregulated by bacteria upon switching from a planktonic to a biofilm lifestyle. Pf4 was subsequently found to form tactoids in the extracellular matrix of biofilms, where it encapsulates bacterial cells, allowing them to tolerate antibiotic treatment[4,15]. Atomic structure of the Pf4 phage capsid, tactoid formation properties and protection from antibiotic treatment was unaffected regardless of whether the genome was still present within the phage filament, showing that the genome did not affect the observed properties of tactoids[4].

The presence of filamentous molecules in a viscous environment is a hallmark of all biofilms[28], however, how these molecules bestow emergent properties to bacteria, including antibiotic protection, is unknown. In this study, we examine tactoid properties, bacterial encapsulation characteristics, and antibiotic protection mediated by two distinct model inoviral phages. We report a 3.2 Å atomic structure of the fd capsid and compare it to Pf4, identifying differing structural and biochemical properties of the two phages. We further show that fd and Pf4 form tactoids with different morphologies and develop a theoretical framework, which shows that morphological differences between tactoids are governed by differences in material properties, which are in turn determined by phage geometry and packing. We show that tactoids of both fd and Pf4 associate with bacterial cells, conferring protection against antibiotic treatment by reducing antibiotic uptake into tactoid-associated cells, showing the formation of a diffusion barrier. Our results explain the factors determining phage tactoid properties, with implications for antibiotic protection provided by filamentous molecules, which are enriched in viscous environments such as bacterial biofilms or sites of infection in hosts.

## Results

### Atomic structure of the fd phage capsid using cryo-EM

To study the biochemical properties of the fd phage capsid, we generated phage preparations using previously established procedures (Methods). We observed ~64 Å wide phage particles on cryo-EM grids (Supplementary Fig. 1a). Using helical reconstruction and applying C5 symmetry, a 3.2 Å-resolution map of the fd phage capsid was obtained, which allowed derivation of an atomic model of the capsid (Fig. 1a–c, Supplementary Figs. 1b–e and 2, Supplementary Movie 1 and Supplementary Table 1). In agreement with previous predictions[29], the atomic model reveals a pentameric (C5) subunit arrangement of the pVIII capsid protein, which forms a single α-helix containing 50 residues. The helix is terminated at the N-terminus of the mature protein by a proline residue (P6), and residues 1–5 of the capsid protein, which are located on the phage surface, are disordered (Fig. 1c). Due to helical symmetry, capsid protein monomers stack almost vertically to form a highly symmetrical arrangement along the helical axis (Fig. 1b). The capsid proteins interact predominantly via a hydrophobic interaction network (Fig. 1d), which include the residue Y21 that was shown previously to induce structural instability[30] and hence mutated in several studies to methionine[11,13,31]. In our specimen of wild-type fd phage, despite retaining Y21, there was no significantly lowered resolution detected at this location in the structure, even though local resolution varied slightly within other parts of the capsid array (Supplementary Fig. 1e).

The C-terminus of the capsid protein, which faces the inner lumen of the cylindrical phage, interacting with the genomic DNA, has four positively charged lysine residues exposed to the lumen (Fig. 1e). These lysines presumably bind to the negatively-charged DNA phosphates, as also seen in other phages[4,5]. While density for the DNA is observed in our fd cryo-EM map, it is not well-resolved and does not support atomic model building of the DNA genome (Supplementary Fig. 1f). Poorly resolved DNA density was also observed in previous studies on class I inoviral bacteriophages (Supplementary Fig. 3), where direct atomic model building into the DNA density could also not be performed[5]. Overall, comparison of our fd capsid structure with previous structural models of fd shows that while the structure of the α-helical capsid subunit pVIII has been well-approximated in some structural models, the arrangement of subunits into the overall capsid architecture was imprecise in all (Supplementary Fig. 4).

### Comparison of Class I fd capsid structure with the Class II phage Pf4

Fd is an archetypal class I inoviral bacteriophage that infects *E. coli*, but does not integrate into its genome[32]. Structural comparison with Pf4[4], a class II inoviral bacteriophage with a prophage lifestyle, shows a slightly larger capsid size (fd: 50 residues, Pf4: 46 residues) and lumen diameter (fd: 24 Å, Pf4: 22 Å, Fig. 2). At the N-termini, both phages contain acidic amino acid residues, leading to a negatively charged outer surface (Fig. 2b, c). In the case of fd, negatively charged residues are mostly located in the disordered N-terminus (sequence AEGDD), while the negatively charged outer surface residues in Pf4 are ordered (Fig. 2c). As a whole, the fd major coat protein is more positively charged than that of Pf4 (compare the isoelectric point of fd 6.28 versus Pf4 4.68), which is due to an increased positive charge facing the inner lumen of the phage with four basic residues present at the C-terminus compared to two in the case of Pf4 (Fig. 2d).

Two arrangements of genomic DNA in inoviruses have previously been proposed. Either the DNA could be linear single-stranded, forming a long helical arrangement, or circular single-stranded DNA, forming a double helix-like arrangement. Initially, all inoviruses were thought to contain circular single-stranded DNA, however, in the case of Pf4, a linear single-stranded-DNA was resolved[4]. In agreement with previous work, our structural model for fd suggests a circular single-stranded arrangement, as the higher positive charge-density of the capsid protein (four basic residues in fd versus two in Pf4) suggests that twice as many negative charges of the DNA phosphate backbone are being stabilised. The length of inoviruses is proportional to their genome length, and the size of the Pf4 genome is about twice as large

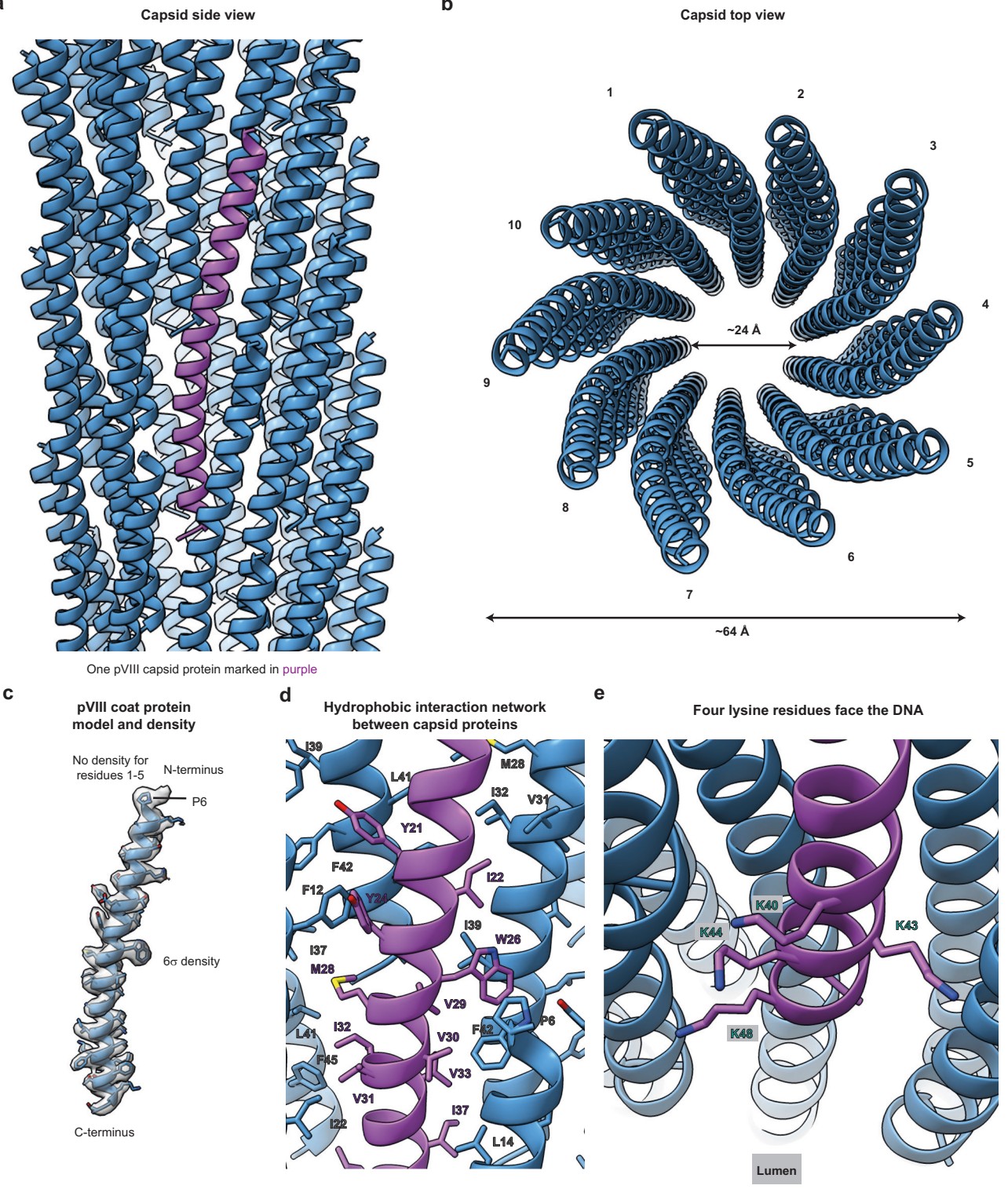

**Fig. 1 | Cryo-EM structure of the fd bacteriophage capsid at 3.2 Å-resolution.**
**a** Side view of the fd capsid (ribbon depiction) with a single pVIII subunit high-
lighted in purple. **b** Top (perspective) view of the capsid shows a 24 Å-wide lumen.
The vertical interactions of capsid proteins results in 10 protofilament-like stacks
observed from the top. **c** Atomic model of pVIII with the cryo-EM density shown as
an isosurface at 6 σ away from the mean value. **d** Hydrophobic interaction net-
works within the capsid, with interacting hydrophobic residues marked. **e** Four
lysine residues near the C-terminus extend into the capsid lumen.

as the fd genome (Pf4: 12.4 kb, fd: 6.4 kb[33]), whilst their phage filament
length differs by about a factor of four (0.9 μm fd[14] vs. 3.8 μm Pf4[4]). This
mismatch suggests that the genome arrangement of fd is different to
Pf4, compacted by a factor of two, supporting a circular arrangement
of its DNA.

To complement our structural studies, we performed atomistic
molecular dynamics (MD) simulations of fd and Pf4 phage capsids to
determine whether the additional positively charged residues in the fd
lumen can interact with higher amounts of negative charges than the
Pf4 capsid. Comparing simulations of fd and Pf4 capsids, performed in

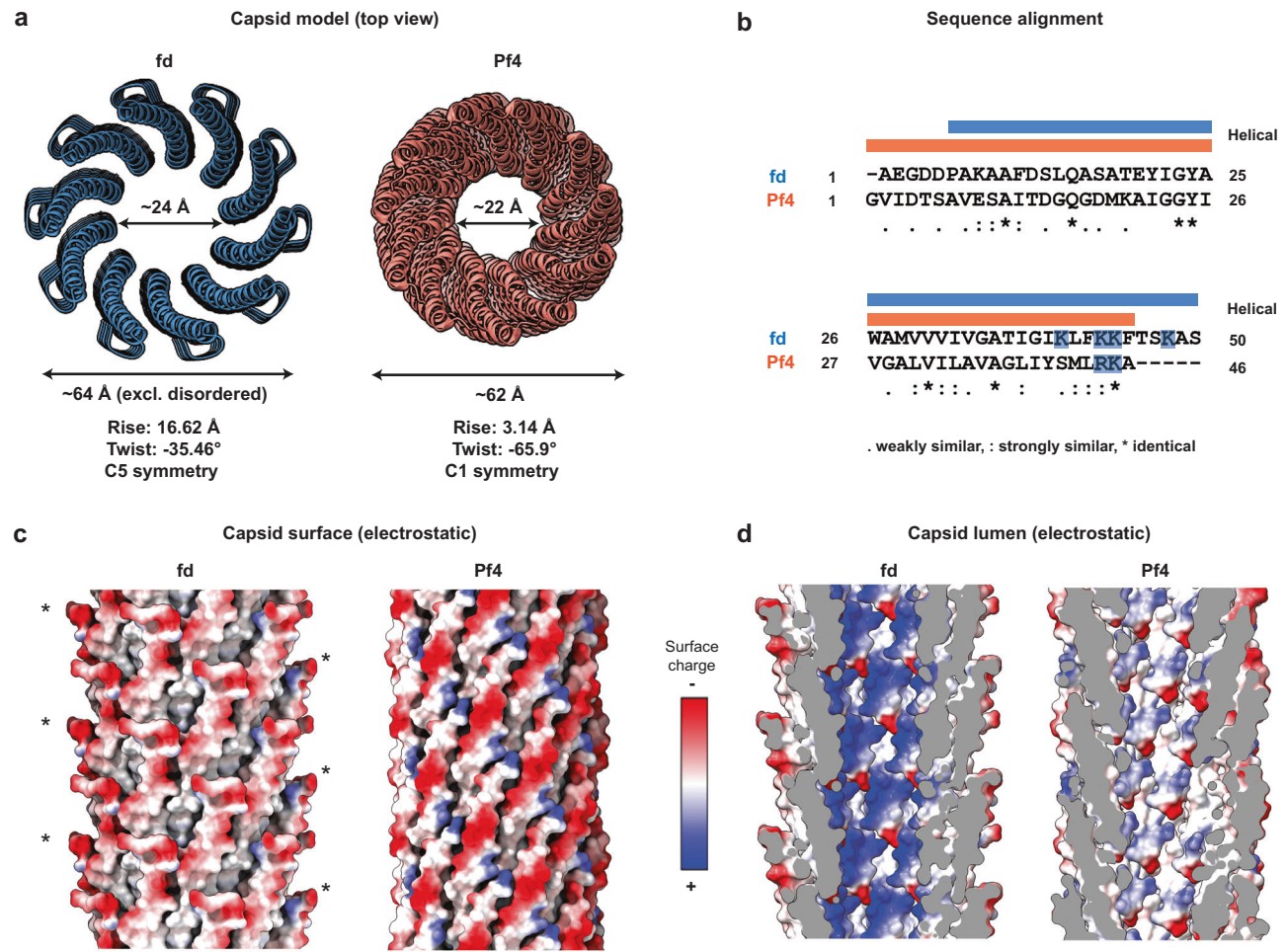

**Fig. 2 | Comparison of the cryo-EM structures of the class I fd phage (this study) and the class II Pf4 phage (PDB 6TUP [https://doi.org/10.2210/pdb6tup/pdb]).** **a** Orthographic top view of fd and Pf4 capsids shown as ribbon diagrams. Flexible residues at the fd N-termini have been modelled to allow comparison with Pf4, where all residues are ordered in the cryo-EM structure. **b** Clustal Omega sequence alignment of the major capsid protein of fd versus Pf4. Basic residues extending into the lumen are marked in blue. Helical regions are marked above, with fd in blue and Pf4 in salmon. **c** Electrostatic surface of the capsid. Disordered residues in fd (1–5) were modelled (*) to allow for a more accurate comparison of electrostatic surfaces. **d** Sliced side view of the capsid lumen depicting electrostatic charge distribution at the luminal surface.

0.15 M salt (NaCl), where both capsids were stable, we found an increased accumulation of negatively charged chloride ions in the fd lumen (containing four lysine residues) compared with Pf4 (containing only two positively charged residues, Fig. 3 and Supplementary Fig. 5). Since our simulations did not contain ssDNA, these increased negatively charged ions in the fd lumen support our expectation from our cryo-EM structures that DNA in the class I inoviral phage fd has a different topology to that in the class II inoviral phage Pf4, which has a linear topology[4]. Also notably, root mean square fluctuation values show flexibility in the N-terminal residues of fd phage (Supplementary Fig. 5), with the N-terminus significantly extending away from the capsid, confirming the flexible nature of this region, in agreement with the lower local resolution detected in our cryo-EM density (Supplementary Fig. 1e).

### Phage geometry and packing govern tactoid morphology

Having resolved the structure of the fd phage capsid and described the biochemical differences with Pf4, we next investigated whether the tactoid-forming properties of both fd and Pf4 are different. We assembled tactoids by mixing each phage with the anionic polysaccharide alginate, a biopolymer commonly found in the extracellular matrix of biofilms[15], and compared both fd and Pf4 tactoids at phage and biopolymer concentrations that gave equivalent levels of total

tactoid formation (Supplementary Fig. 6a–c). Under these conditions, we found that individual fd tactoids have a different morphology, evidenced by a significantly higher volume and smaller aspect ratio (ratio of the major to minor elliptical axis) than individual Pf4 tactoids (Fig. 4a–c and Supplementary Fig. 6). Replacing alginate with the glycosaminoglycan hyaluronan, a common biopolymer found on epithelial cells relevant in *P. aeruginosa* infections, resulted in equivalent tactoid morphologies (Supplementary Fig. 7). Interestingly, electron cryotomography (cryo-ET) of small tactoids amenable to imaging shows that fd tactoids consist of laterally-packed phages, with Fourier transforms showing a regular spacing of ~60 Å between phages. This is in contrast to tactoids of Pf4 showing weaker lateral ordering, but nevertheless tight overall packing, with no discrete peaks in the Fourier transforms (Fig. 4d, e, Supplementary Movie 2). Despite differences in packing, phages were found to show similar mobility within both fd and Pf4 tactoids as shown by FRAP experiments (Supplementary Fig. 8, Supplementary Movie 3). Interestingly, however, the immobile fraction of Pf4 was found to be considerably higher than for fd (13.9% for fd vs 45.6% for Pf4; Supplementary Fig. 8), which suggests that these tactoids, despite exhibiting local liquid crystalline order, show differing degrees of molecular exchange.

While it has previously been established that phage tactoids assemble due to depletion attraction interactions[15], it is unclear why

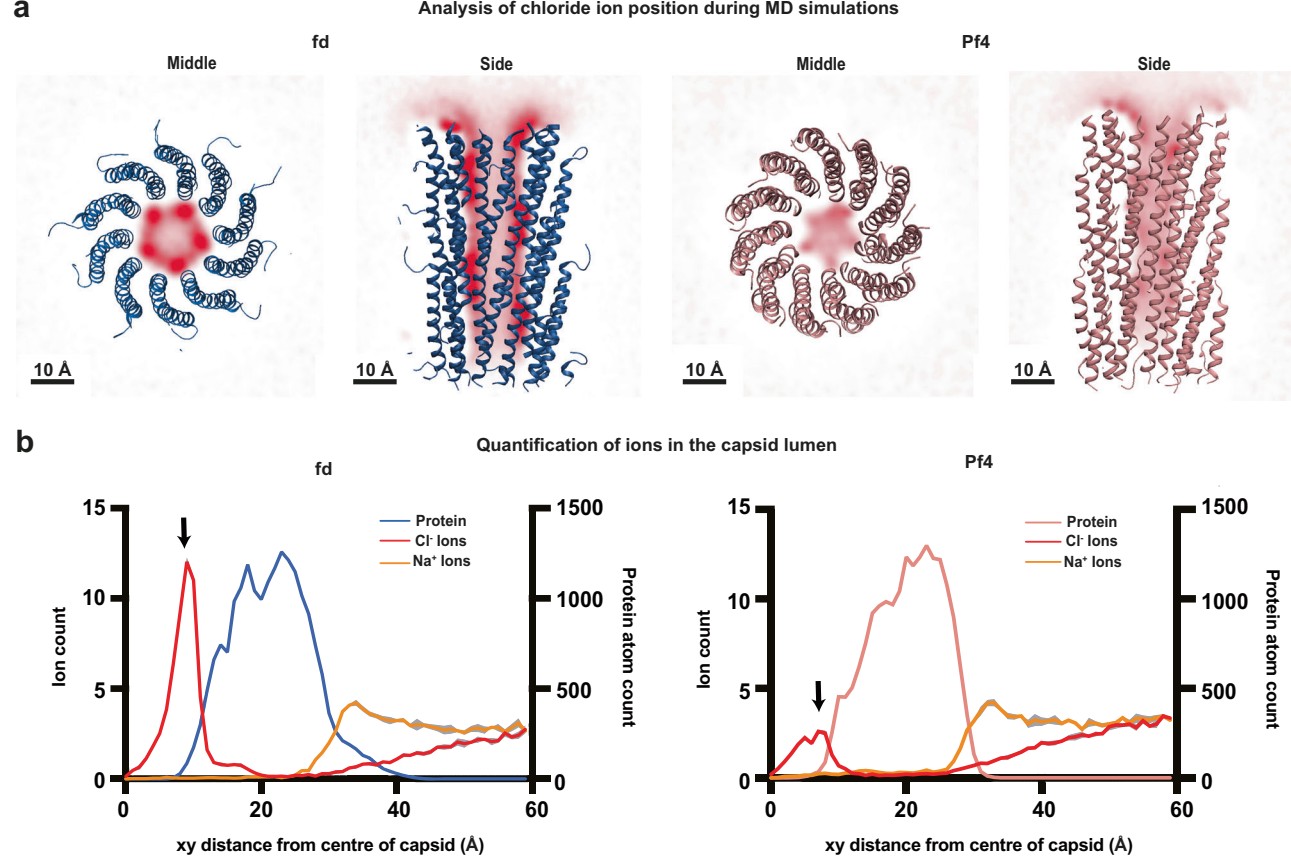

**Fig. 3 | Atomistic molecular dynamics simulation analysis of fd and Pf4 capsids. a** Weighted ionic density of chloride ions (computed using VMD volmap) averaged over all trajectory frames. Density of chloride ions is shown in red. Dark red indicates higher ionic density. Fd capsid (left), shown in blue, depicts high levels of chloride ions in the capsid lumen (sliced at the midpoint of the filament). Pf4 capsid (right), shown in salmon, has comparatively lower levels of chloride ions in the capsid lumen (sliced at the midpoint of the filament). Both systems were simulated in 0.15 M NaCl. **b** Quantification of ion number at different positions in a

cross section of the phage, starting from the centre of the capsid for fd (left) and Pf4 (right). Fd protein atoms are coloured blue, Pf4 coloured salmon, chloride ions red and sodium ions yellow. Protein atoms are shown for clarity. The recruitment of chloride ions to the interior of the capsid can be seen as a peak at approximately 10 Å (indicated by arrows), with a higher peak observed for fd as compared to Pf4. Mean of four repeats (simulations) is plotted for each system in bold colour, with the standard error of the mean (SEM) plotted in grey. See Supplementary Fig. 5 for additional analyses. Source data for graphs are provided as a Source Data file.

the shape of the tactoids differs between fd and Pf4. We hypothesised that the considerable difference in phage geometry and size (lengths 0.9 μm for fd vs 3.8 μm for Pf4) is the cause of differences in tactoid morphology. To understand which phage properties governed the observed tactoid morphologies, we developed a physical model of tactoids containing hard rods, to link the geometrical properties of the phages to those of the tactoids. First, we performed a theoretical scaling calculation[34], involving minimising the free energy of a tactoid that accounts for surface (interfacial) and volumetric (elastic) contributions (see theory section in Methods for details). In agreement with previous studies[34-36], the scaling calculation predicts the following relationship between the geometrical properties of the tactoid and its physical properties:

$$\frac{R}{r} \propto \left(\frac{K}{\gamma}\right)^{3/5} V^{-1/5}, \qquad (1)$$

where R is the long (major) axis of the tactoid, r is the short (minor) axis of the tactoid, K is the Frank elastic constant, γ is the surface tension, and V is the tactoid volume (see Fig. 4f for a schematic of the model). The relation, which is valid for $r \ll R$, predicts a decrease in the aspect ratio of the tactoid with an increase in the size (volume) of the tactoid. As predicted, through lines of best fit to $R/r = CV^{-1/5}$, where C is the fitting parameter $C_{Pf4} = 5.88 \pm 0.11 \ \mu m^{3/5}$ and $C_{fd} = 3.72 \pm 0.09 \ \mu m^{3/5}$,

we observed that the aspect ratios and volumes of both the fd and Pf4 tactoids displayed this relationship (Fig. 4f, g). Interestingly, we found the aspect ratios of Pf4 tactoids to be approximately $C_{Pf4}/C_{fd} \sim 1.6$ times larger than the aspect ratios of the fd tactoids at similar volumes (Fig. 4g).

To explain the observed difference between the Pf4 and fd curves in Fig. 4g, we performed a second scaling calculation to link the material tactoid properties in the prefactor of Eq. (1) to the phage geometry and packing within the tactoid (see Methods). By approximating the elastic constant K and the surface tension γ (see Eq. (7), Methods), we derived the following scaling relationship:

$$\frac{R}{r} \propto \left(\rho \frac{b^2}{a}\right)^{3/5} V^{-1/5}, \qquad (2)$$

where ρ is the packing fraction of the phages in the tactoid ($\rho_{fd} = 0.9$ and $\rho_{Pf4} = 0.25$, as determined from analysis of tomograms; see Methods), b is the length (major axis) of the phage ($b_{fd} = 0.9 \ \mu m$ and $b_{Pf4} = 3.8 \ \mu m$), and a is the width (minor axis) of the phage ($a_{fd} = 6$ Å and $a_{Pf4} = 6$ Å). Surprisingly, given the highly coarse-grained nature of the model, Eq. (2) predicts $C_{Pf4}/C_{fd} \sim 2.6$, a value which is within a factor of two from the fits to the experimental data (Fig. 4g). We conclude that Eq. (2) contains the key determinants of tactoid shape: the shape

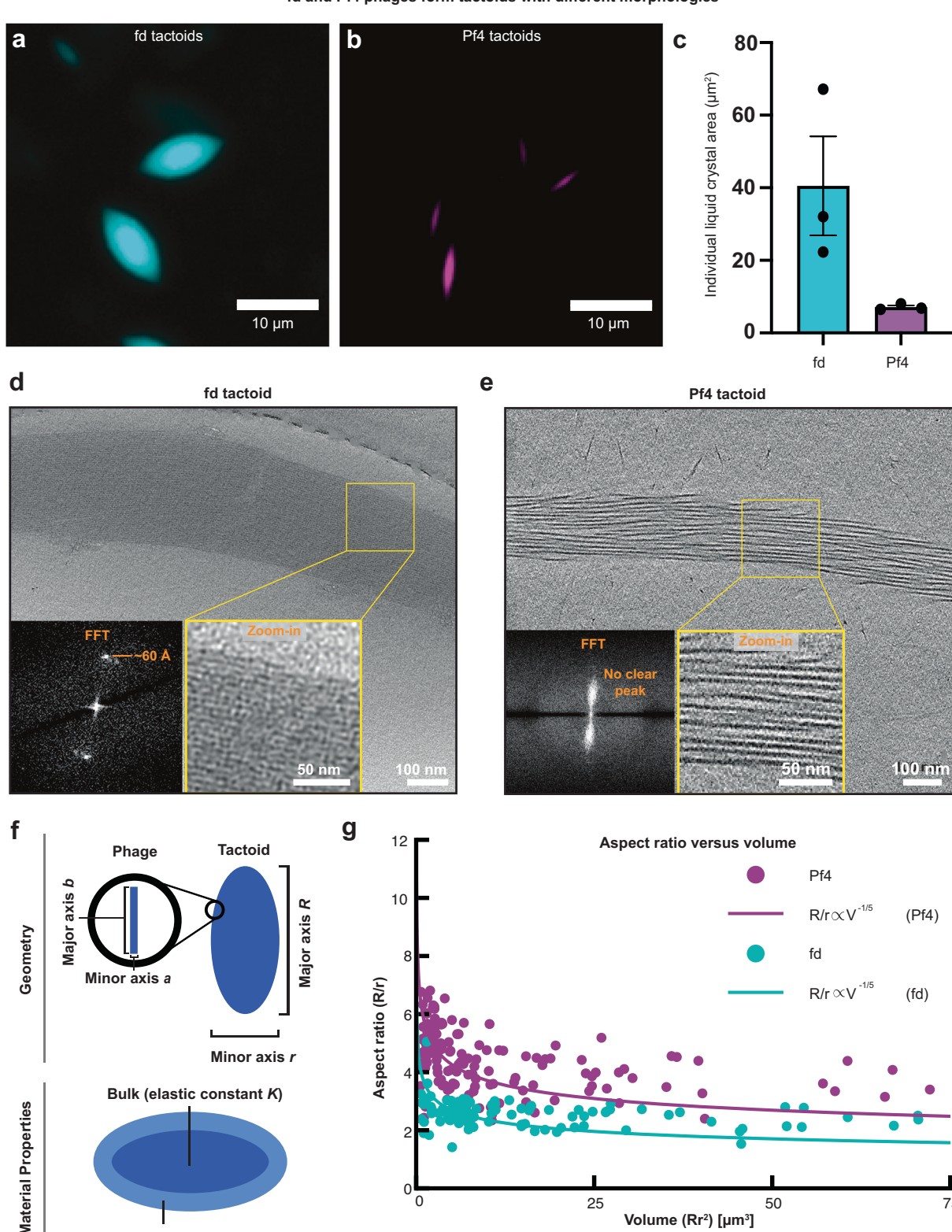

**Fig. 4 | Comparison of fd and Pf4 tactoid morphology and filament packing.**
**a**, **b** Representative light microscopy images of tactoids formed by Alexa-488-labelled fd (cyan) and Pf4 phages (magenta). **c** Bar chart showing the average area of individual tactoids as assessed by light microscopy followed by segmentation of tactoids. Values shown are the mean of three independent experiments and error bars represent standard deviation. **d**, **e** Tactoid morphology as observed via cryo-ET. Inset: Zoom-in and Fourier Transform. Images are representative of 3 tomograms and 10 tomograms respectively. **f** Schematic of the coarse-grained model wherein phages are modelled as hard rods and the phage tactoids are modelled as tactoids with bulk (elastic) and surface energetic contributions (see Eqs. (2–3)). **g** Tactoid aspect ratio as a function of tactoid volume. Tactoids, as visualised via light microscopy, follow the scaling law $R/r \propto V^{1/5}$ as shown through lines of best fit to $R/r = CV^{1/5}$, where $C$ is the fitting parameter ($C_{Pf4} = 5.88 \pm 0.11$ $\mu m^{3/5}$ and $C_{fd} = 3.72 \pm 0.09$ $\mu m^{3/5}$). Source data for graphs are provided as a Source Data file.

and packing of the phages, and the overall size of the tactoid. The small quantitative discrepancy between the predicted and measured values of $C_{Pf4}/C_{fd}$ could be caused by differences in specific phage-phage interactions (e.g., electrostatic surface interactions; see Fig. 2) or through distinct phage geometry causing differing depletion interactions. While these interactions already feed into Eq. (2) through the phage packing fraction $\rho$, such interactions also affect the material properties $K$ and $\gamma$ in Eq. (1), and would therefore enter Eq. (2) as prefactors. However, the reasonably good agreement between experiment and theory suggests that differences in $K$ and $\gamma$ between Pf4 and fd tactoids are mostly determined by phage aspect ratio. Taken together, our experimental measurements and scaling theory suggest that overall tactoid morphology is governed by biophysical effects through tactoid size and material properties, which are determined by phage geometry and packing.

## Tactoid encapsulation of bacterial cells

Since generic biophysical effects (shape and packing) appear to govern overall tactoid morphology, we next asked how these morphologically distinct tactoids interact with rod-shaped *E. coli* and *P. aeruginosa* bacteria. Previous studies have shown that tactoids formed by Pf4 can fully encapsulate *P. aeruginosa* cells, which correlates with protection from antibiotic treatment[4]. However, it is unclear whether this encapsulation is driven by specific chemical interactions between the phage and the cell surface, or whether it may be driven by physical factors. To answer this question, we mixed fd and Pf4 tactoids with *E. coli* or *P. aeruginosa* cells, which are bacteria with significantly different surface chemistry due to different surface proteomes and the lack of a lipopolysaccharide O-antigen in *E. coli* K12[37]. Pf4 tactoids could associate well with both *P. aeruginosa* (its native host) and *E. coli* (non-host) cells (Fig. 5b, f; Supplementary Fig. 9). Interestingly, fd tactoids also associated closely with both *P. aeruginosa* (non-host) and *E. coli* (native host) cells (Fig. 5a, e; Supplementary Fig. 9). This observation suggests that the interaction is not determined by specific phage-bacteria interactions, but rather the biophysical phage and tactoid properties.

Curiously, although both phage tactoids could associate with bacterial cells, Pf4 tactoids more readily encapsulated either bacterium and were able to completely surround the cell, whereas fd tactoids interacted with cells laterally (Fig. 5). Using semi-automated analysis of multiple tactoid:cell images, we measured the pairwise orientation differences between the long axes of bacterial cells and their associated tactoids (Fig. 5c, g). Our measurements show an alignment of axes in all cases, with the tightest alignment between Pf4 tactoids and *P. aeruginosa* (8.9 ± 9.2°; significantly different from Pf4 with *E. coli* 12.3 ± 13.1°, $P_{value} < 0.001$). The orientational difference of fd tactoids with both bacteria was larger, but still alignment was observed, suggesting the interaction is not stochastic (fd with *P. aeruginosa* 13.3 ± 13.1°, fd with *E. coli* 14.6 ± 13.9°; no statistically significant difference) (Fig. 5c, g). Fd tactoids associated with bacteria were also larger than Pf4 tactoids associated with both bacteria tested, with fd tactoids ~2.5–2.9 μm larger compared to Pf4 tactoids ~1.2–1.4 μm larger than the bacterial cell major axis (Supplementary Fig. 9a–d).

We wondered whether the observed qualitative differences in encapsulation of bacterial cells – that Pf4 tactoids more readily encapsulated the bacteria as compared to fd tactoids (Supplementary

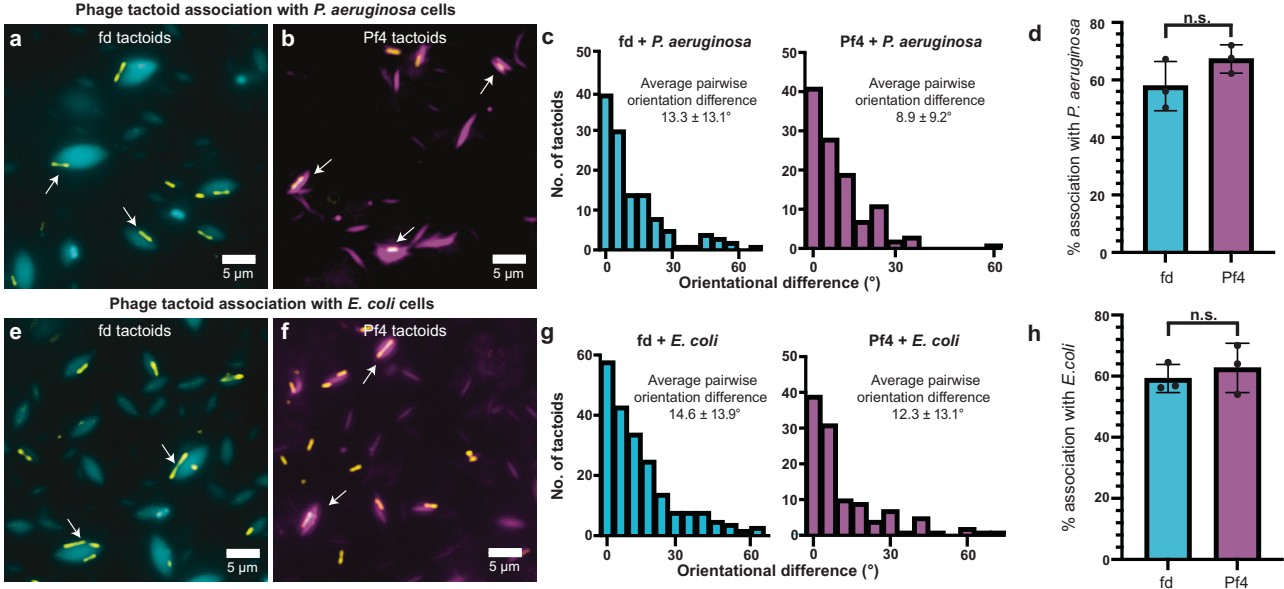

**Fig. 5 | Comparison of fd and Pf4 tactoid association with bacterial cells.**
**a**, **b** Representative images showing association of fd and Pf4 tactoids with *P. aeruginosa* cells observed in light microscopy. Transmitted light channel shows bacteria (yellow) and fluorescence channel shows fd (cyan) and Pf4 (magenta) tactoids, respectively. **c** Histogram of pairwise orientational differences between bacterial cells and associated tactoids from semi-automated segmentation of images. Values reported are the mean and standard deviation from three independent experiments. Left panel fd + *P. aeruginosa* (n = 144), right panel Pf4 + *P. aeruginosa* (n = 163), significantly different with ***$P_{value}$ = 0.0010. **d** Percentage of *P. aeruginosa* cells associated with tactoids (n = 205 for fd + *P. aeruginosa* and n = 216 for Pf4 + *P. aeruginosa*). Differences in association are not statistically significant, $P_{value}$ = 0.1735. Values shown are the mean of three independent experiments and error bars represent standard deviation. **e**, **f** Corresponding representative images showing association of fd and Pf4 tactoids with *E. coli*. Transmitted light channel shows bacteria (yellow) and fluorescence channel shows fd (cyan) and Pf4 (magenta) tactoids, respectively. **g** Histogram of pairwise orientational differences between bacterial cells and associated tactoids from semi-automated segmentation of images. Values reported are the mean and standard deviation from three independent experiments. Left panel: fd + *E. coli* (n = 244), right panel: Pf4 + *E. coli* (n = 137), no significant difference, $P_{value}$ = 0.1190. Pf4 + *P. aeruginosa* versus Pf4 + *E. coli*, *$P_{value}$ = 0.0110. Fd + *P. aeruginosa* versus fd + *E. coli*, no significant difference, $P_{value}$ = 0.3623. Pf4 + *P. aeruginosa* versus fd + *E. coli*, ***$P_{value}$ < 0.0001. Fd + *P. aeruginosa* versus Pf4 + *E. coli*, no significant difference, $P_{value}$ = 0.5404. **h** Percentage of *E. coli* cells associated with tactoids (n = 344 for fd + *E. coli* and n = 188 for Pf4 + *E. coli*). Differences in association are not statistically significant, $P_{value}$ = 0.5519. Values shown are the mean of three independent experiments and error bars represent standard deviation. All *p*-values were calculated using an unpaired *t* test. Source data for graphs are provided as a Source Data file.

Fig. 9) – could be explained by biophysical phage and tactoid properties within our coarse-grained framework. Based on our approximations of the bulk elastic constant $K$ and surface tension $\gamma$ in terms of the phage packing fraction $\rho$ and phage shape (width $a$ and length $b$), we predict that Pf4 tactoids have a ~1.2 times larger $K$ but ~4 times smaller surface tension $\gamma$ than fd tactoids. By considering the bacteria as the wetting surface for the phages, the Young-Laplace equation for the interior contact angle θ (ref. 38) gives

$$\cos(\theta) = \frac{\gamma_{b-d} - \gamma_{b-a}}{\gamma}, \tag{3}$$

where $\gamma_{b-d}$ and $\gamma_{b-a}$ are the respective bacteria-tactoid (assumed constant) and bacteria-alginate (assumed constant) surface tensions, and $\gamma$ is the alginate-tactoid surface tension already defined above. Equation (3) suggests that a higher alginate-tactoid surface tension $\gamma$ results in an increase in contact angle between the tactoid and bacteria interfaces, i.e., reduced wetting. Therefore, we predict reduced wetting (and therefore reduced encapsulation) of bacteria by fd tactoids because of their higher surface tension than Pf4 tactoids. This is in line with experiments (Fig. 5), further confirming that material properties, which are determined by phage shape and packing, drive the observed differences in behaviour between Pf4 and fd tactoids.

### Tactoid-mediated antibiotic protection

In our previous work, we showed that encapsulation was tightly linked with antibiotic protection[4]. To test how the observed differences in association of tactoids with cells affected antibiotic protection, we used a previously developed antibiotic protection assay[4] that measures bacterial survival in different conditions against an antibiotic challenge. Our assay showed that despite clear and quantifiable differences between fd and Pf4 tactoid association with cells (Fig. 5 and Supplementary Fig. 9), both fd and Pf4 tactoids could protect *P. aeruginosa*, corroborating previous reports[4,15], with Pf4 tactoids providing higher levels of protection than fd tactoids. We further found that the presence of either tactoid could also protect *E. coli* bacteria from antibiotic challenge (Fig. 6).

To probe the link between tactoid association and protection from antibiotics, we next measured antibiotic uptake of *P. aeruginosa* cells in the presence and absence of tactoids using optical microscopy. To this end, a fluorescently-labelled antibiotic (Texas-Red gentamicin) was added to *P. aeruginosa* cells in the presence or absence of phage tactoids. Antibiotic uptake into cells was then followed using the Texas Red fluorescent signal. In a control sample lacking tactoids, significant antibiotic accumulation could be measured within the cells. In contrast, bacteria encapsulated by tactoids had significantly reduced internalised antibiotic compared to the control, while non-encapsulated bacteria in the same sample showed significantly higher uptake than encapsulated cells (Fig. 6c–k and Supplementary Fig. 10). The same decrease in antibiotic uptake was observed when Pf4 ghost phage, chemically treated with lithium chloride to remove phage DNA was used (Fig. 6i–k), suggesting this effect is independent of the presence of the genome within the phage. This supports a model where tactoids create a diffusion barrier around the cell that prevents the antibiotic from reaching the cell surface, inducing tolerance. Taken together, our results suggest that biophysical phage and tactoid properties profoundly affect the assembly of filamentous phages into tactoids. These parameters also govern the association of tactoids with bacterial cells, thus influencing bacterial tolerance to antibiotic treatment.

### Discussion

In this study, we have solved the capsid structure of the intensely studied, archetypal class I inoviral bacteriophage fd. Being the first discovered inovirus, fd is used in biotechnology for phage display[9] as well as for investigating phase transitions of colloidal rod-like molecules[10]. Numerous inconsistent models for the fd phage capsid have been proposed over the years by X-ray fibre diffraction, ssNMR and 8 Å-resolution cryo-EM[11–14,29,39–41]. While lack of resolution in the past studies probably led to these inconsistencies, the advent of improved cryo-EM technology, including improved microscope optics, detectors and image analysis software[42], allowed us to produce a 3.2 Å-resolution cryo-EM structure of the fd capsid. This settles the long-standing debate over fd capsid structure and shows how individual capsid monomers form a cylindrical phage with five-fold symmetry (Fig. 1). Also notably, previous studies had suggested that the Y21 residue within the pVIII capsid protein induces structural flexibility, and that the mutation Y21M could reduce this flexibility[30]. Our structure suggests that while Y21 is involved in hydrophobic interactions within the capsid, it does not act as a structural hinge, evidenced by our high-resolution map. While previous cryo-EM studies had suggested there may be a structural continuum of fd structures[14], we did not observe these states in our analyses.

Comparing the class I fd capsid structure with the class II Pf4 phage reveals four lysine residues extending into the lumen (compared with two positively charged residues in Pf4), where they could interact closely with ssDNA (Fig. 2). While density for DNA could be detected in our map (Supplementary Fig. 1f), it did not allow for unambiguous atomic model building, consistent with previous studies on the Ike phage[5] (Supplementary Fig. 3). The size of the fd phage lumen, its increased positive charge, along with our MD simulations (Fig. 3), suggest that the DNA arrangement in this phage is markedly different to Pf4. Previous literature suggested that fd contains a circular single-stranded DNA genome[29], and this is supported by our data as the linear ssDNA in Pf4 would require less positively charged residues for encapsidation. Given that fd has a comparable number of capsid proteins per length unit, the fact that the fd capsid can compensate twice as many negative charges as Pf4 suggests that its DNA genome indeed is circular. Further studies should probe the organisation of inoviral phage DNA genomes, as demonstrated previously for RNA viruses using Monte Carlo simulations[43], which could provide important insights into the organisation of the DNA genome and its interaction with the capsid.

Fd and Pf4 form tactoids with starkly differing morphology at the same overall specimen tactoid volumes. We found that individual fd tactoids had a smaller aspect ratio and significantly higher volume than Pf4 tactoids, which were smaller and elongated, with a differing phage packing within the tactoid at equivalent conditions (Fig. 4). According to our theoretical calculations and predictions, differences in tactoid material properties, which are determined by phage shape and packing fraction, are sufficient to explain the differences in emergent tactoid morphology. Interestingly, FRAP experiments show a higher immobile fraction of Pf4 compared to fd. This is despite tighter packing of phages in fd tactoids, together underlining the differing material properties between fd and Pf4 tactoids. While factors influencing packing of rods within tactoids are not completely understood, these results together indicate that the biophysical characteristics of the phage dominate in the formation of tactoids. Similar biophysical governance was also observed in tactoid association with cells. Pf4 tactoids almost fully encapsulated rod-shaped *P. aeruginosa* and *E. coli* cells, compared to fd tactoids, which did not fully encapsulate cells, but nevertheless closely associated with bacteria (Fig. 5). Combined with previous studies[4,15], our results suggest the intriguing possibility that the material properties of the phage and tactoid – namely, phage shape and packing, which determine tactoid surface tension and elastic constant, have been evolutionarily selected to maximise protective effects. Future studies could employ genomically engineered phages with various lengths to further probe the role of phage material properties on tactoid formation and encapsulation. Furthermore, mutation of the surface charges could enlighten how far significant

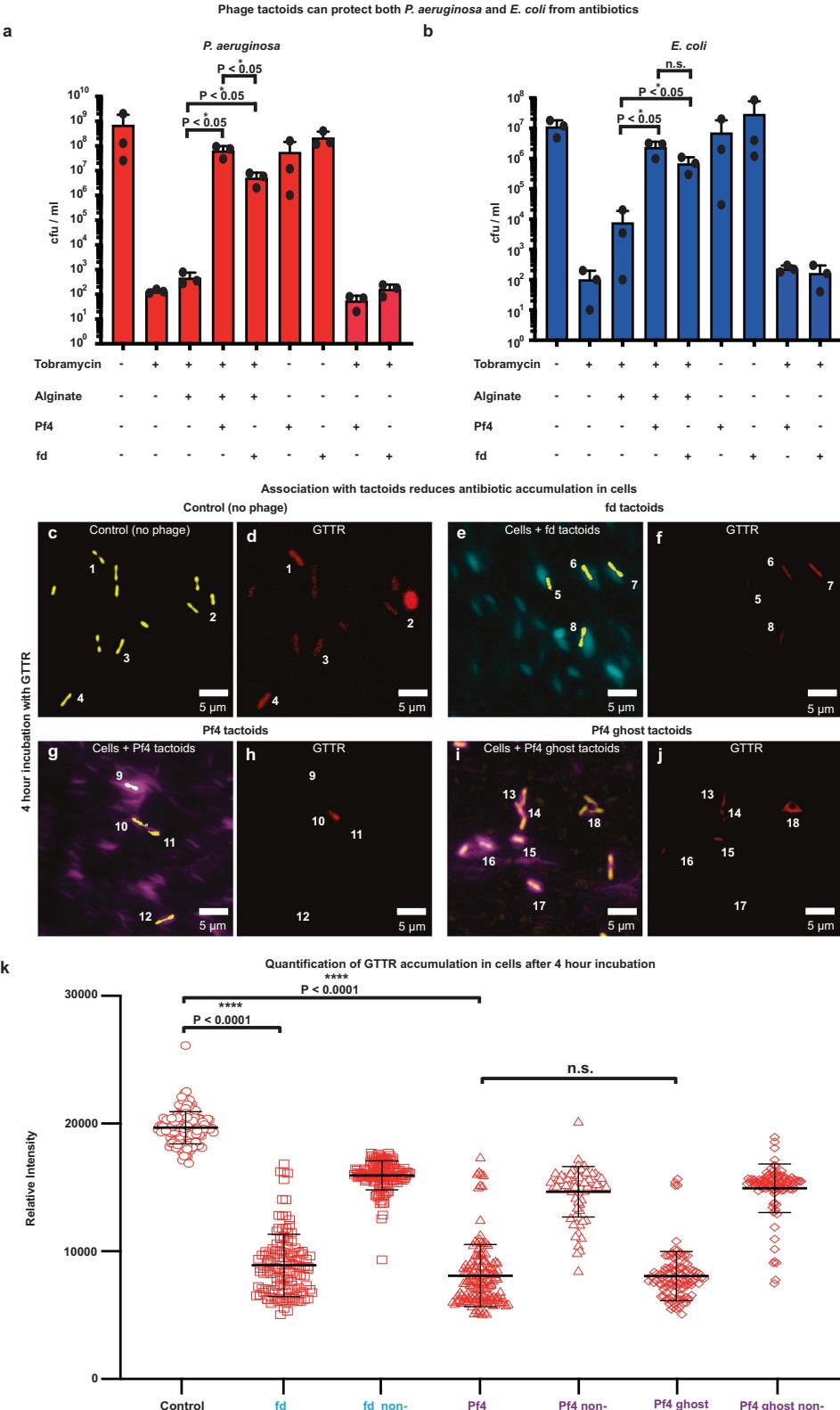

alteration of the phage surface would perturb the biophysical effects described in this study. In addition, simulation techniques could advance our understanding of phage tactoid interaction with bacterial cells. Previous studies have utilised molecular simulations and theoretical analyses to study the interactions of surfactants to various interfaces, enabling the determination of contact angles and line tensions[44,45]. Such methods may provide further mechanistic insights into tactoid association with bacterial cells.

Curiously, tactoid association with cells was sufficient for antibiotic protection (Fig. 6), rather than complete encapsulation. Even though fd tactoids do not fully encapsulate cells, association with tactoids still provided protection against antibiotic treatment to rod-

**Fig. 6 | Association with tactoids protects bacterial cells from antibiotic uptake, suggesting the formation of a diffusion barrier.** Bar graphs showing viability of cells in colony-forming units (cfu) per ml (y-axis) in the presence of different reagents (x-axis) from three independent experiments for (**a**) *P. aeruginosa* and (**b**) *E. coli* against tobramycin treatment. Both Pf4 and fd tactoids protect *P. aeruginosa* to a significantly greater extent than sodium alginate alone ($P_{value}$ = 0.0249 and 0.0384 respectively). Fd tactoids protected significantly less well than Pf4 tactoids ($P_{value}$ = 0.036). Both Pf4 and fd tactoids protect *E. coli* to a significantly greater extent than sodium alginate alone ($P_{value}$ = 0.0271 and 0.0401 respectively). No significant difference (n.s.) was observed between the level of protection provided by fd and Pf4 tactoids ($P_{value}$ = 0.0830). **c–j** Fluorescence and light microscopy images of Alexa 488-labelled fd, Pf4 and Pf4 ghost phage tactoids incubated with Texas Red-labelled gentamicin (GTTR) for 4 h. Shown are Alexa488 phage signal (Cyan – fd, Magenta – Pf4) and pseudocoloured cells (yellow) as determined through brightfield light microscopy (left), and Texas Red signal

corresponding to uptake of the fluorescently labelled antibiotic GTTR by cells (right). Numbering indicates site of cells in corresponding images. Images are representative of 30 images taken over three biological replicates (Control, $n$ = 149 cells, fd associated, $n$ = 128, fd non-associated, $n$ = 128, Pf4 associated, $n$ = 145, Pf4 non-associated, $n$ = 69, Pf4 ghost associated, $n$ = 102 and Pf4 ghost non-associated, $n$ = 100). **k** Plot quantifying GTTR uptake after 4 h respectively in conditions indicated on the x-axis. Bar shows the mean of three independent experiments and error bars represent standard deviation. Association with both Pf4 and fd tactoids results in significantly reduced antibiotic uptake as compared to a control with no phage (fd $P_{value}$ < 0.0001, Pf4 $P_{value}$ < 0.0001), and as compared to cells in the same sample that are not tactoid-associated (fd $P_{value}$ < 0.0001, Pf4 $P_{value}$ < 0.0001). No significant difference (n.s.) in antibiotic uptake was observed between Pf4 and Pf4 ghost tactoids ($P_{value}$ = 0.9040). All *p*-values were calculated using an unpaired *t* test. Source data for graphs are provided as a Source Data file.

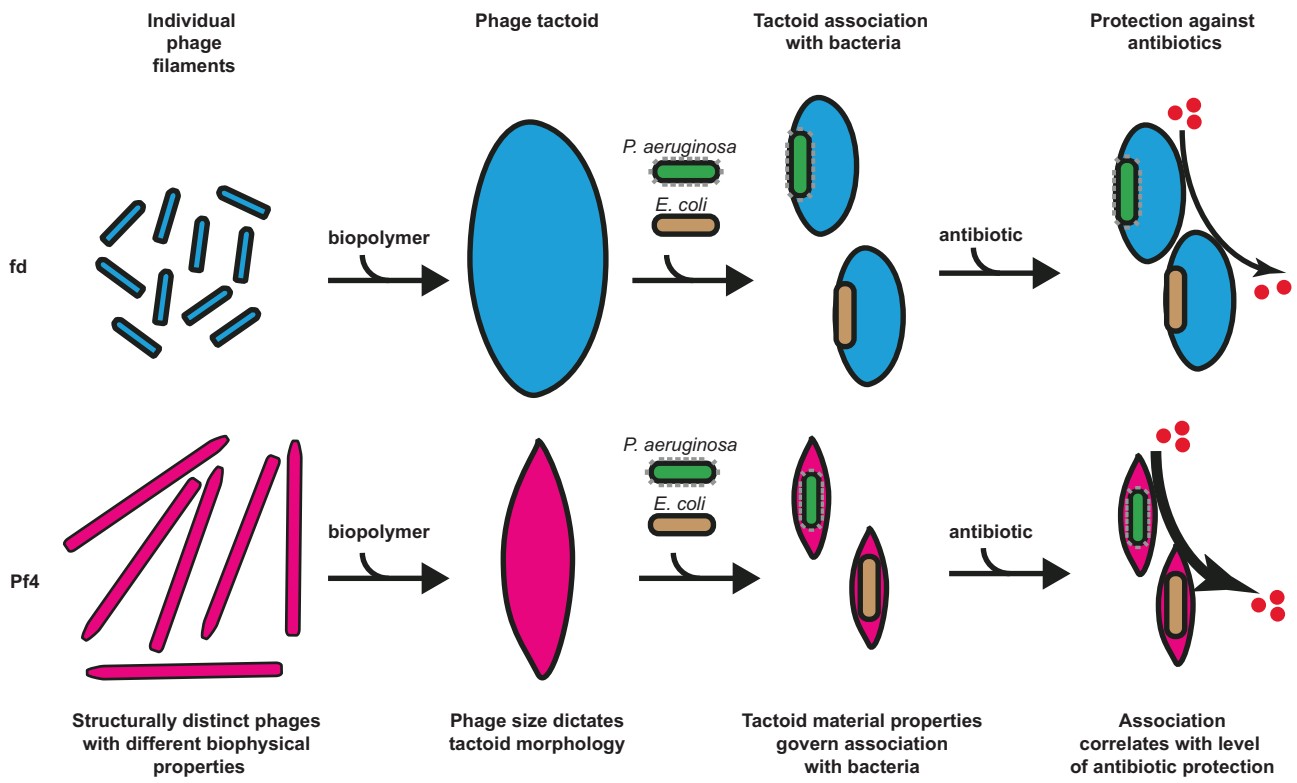

**Fig. 7 | Schematic depicting biophysical nature of phage tactoid-mediated antibiotic tolerance of bacteria.** The inoviruses, fd and Pf4, form tactoids with liquid-crystalline order in the presence of biopolymer with distinct morphologies dictated by phage size. Both fd and Pf4 phage tactoids associate with *P. aeruginosa* and *E. coli* K12, which have different cell surface chemistries. The material

properties of the tactoids govern the association with bacteria independent of their cell surface properties. Phage tactoid association with bacteria results in a diffusion barrier that leads to increased antibiotic tolerance, which correlates with the level of bacterial cell encapsulation by the tactoid.

shaped bacteria (Fig. 6). We have previously proposed that the viscous surrounding created by tactoids leads to reduced antibiotic diffusion and reduced access of molecules to the bacterial cell, conferring protection[4]. Indeed, the use of fluorescently labelled antibiotics shows that encapsulated cells show reduced antibiotic uptake compared to non-encapsulated cells (Fig. 6). This experiment suggests that by limiting the accessible surface of bacteria, tactoids form a diffusion barrier that mediates protection from antibiotics. This effect was independent of whether the DNA genome was still present within the phage filaments, suggesting that while the genome plays a crucial role in phage assembly and lifecycle, its absence does not alter diffusion of the antibiotic.

Since bacteria often proliferate in environments rich in rod-like or filamentous molecules accompanied by biopolymers, for example in the biofilm matrix[4,15,46] or on tissues covered with mucus[47], our data suggest a paradigm where the geometric properties of filamentous components govern assembly into higher-order structures with corresponding material properties that shield cells from antimicrobials, imparting tolerance (Fig. 7). Previous studies have suggested that entire biofilms could be considered to be a nematic, liquid-crystalline system, including alignment of cells with a high three-dimensional order[48,49]. Our research on a minimal system consisting of cells and tactoids shows how filamentous molecules assemble into liquid crystalline structures that can mediate the emergent biofilm property of

antibiotic tolerance. Future research including more biochemically reconstituted components of the biofilm extracellular matrix will be needed to unambiguously prove this proposal and to delineate the exact contribution of each component in biofilm formation. Since filamentous molecules are abundant in most bacterial biofilms[50], the biophysical mechanisms reported in this study could be widespread across bacteria.

## Methods

### Bacterial strains and growth conditions

*E. coli* strains ER2738 (a kind gift from Prof. Eric Grelet, Centre de Recherche Paul Pascal) and XL1 (a derivative of the K12 strain) were used for fd phage amplification and encapsulation assays respectively. *P. aeruginosa* strains PAO1 or PAO1 Δ*PAO728* (a kind gift from Prof. Patrick Secor, University of Montana) were used for experiments as indicated. Unless otherwise specified, shaking cultures of bacteria were incubated at 37 °C in Luria-Bertani (LB) medium with agitation at 180 rpm (revolutions per minute).

### Phage production and purification

Fd was amplified from a starter preparation of purified fd (a kind gift from Prof. Eric Grelet, Centre de Recherche Paul Pascal) using ER2738 *E. coli*. Briefly, to propagate fd phage, a 250 ml culture of ER2738 *E. coli* at $OD_{600}$ 0.3–0.4 was infected with 100 μl of fd phage at 1 mg/ml and grown overnight at 37 °C with agitation. Cells were pelleted by centrifugation (5000 g, 20 min, 4 °C) and the supernatant further centrifuged to remove residual cells (15,000 g, 30 min, 4 °C). The supernatant was adjusted to 0.5 M NaCl and phage precipitated overnight with 10% (w/v) PEG (polyethylene glycol) 8000. Precipitated phage was harvested by centrifugation (12,000 g, 30 min, 4 °C). The phage containing pellet was resuspended in PBS (phosphate buffered saline) and dialysed against PBS overnight using 10 kDa MWCO (molecular weight cut-off) snakeskin dialysis membranes (Thermo-Fisher). Pf4 was amplified from a stock obtained from static *P. aeruginosa* PAO1 biofilms. For amplification of Pf4, $1 \times 10^3$ pfu (plaque forming units)/ml of the Pf4 isolated from PAO1 biofilms were incubated with 1 ml of PAO1 culture at 0.5 $OD_{600}$ for 15 min and mixed with hand-hot 0.8% (w/v) agar. The mix was plated onto 10 cm² LB-agar plates and incubated overnight at 37 °C. Each plate was covered with 5 ml PBS and incubated for 6 h before the PBS was collected, centrifuged (12,000 g, 30 min, 4 °C) and the supernatant adjusted to 0.5 M NaCl and phage precipitated with 10% (w/v) PEG 6000. Precipitated phage was harvested by centrifugation (12,000 g, 30 min, 4 °C). The phage containing pellet was resuspended in PBS and dialysed against PBS overnight using 10 kDa MWCO snakeskin dialysis membranes (ThermoFisher). Yield of both fd and Pf4 phage preparations was estimated using Nanodrop (Thermo Scientific). Pf4 ghosts, lacking the DNA genome, were prepared by treating Pf4 (5 mg/ml) with 10 M lithium chloride in a 1:1 (v/v) ratio at 46 °C for 2 days. The sample was then treated with DNaseI (10 μg/ml) and benzonase (1 μ/ml) at 37 °C for 2 h. The sample was centrifuged to pellet phage at 100,000 x*g* for 1 h at 4 °C and the subsequent pellet resuspended in PBS.

### Fluorescent labelling of phage

Purified fd or Pf4 or Pf4 ghost phage was dialysed into 10 mM sodium carbonate buffer pH 9.2 using a 10 kDa MWCO snakeskin dialysis membrane (ThermoFisher). One ml of fd or Pf4 phage (5 mg/ml) was incubated with 100 μg A488 fluorescent dye (ThermoFisher) for 1 h at room temperature (RT) with end-over-end agitation. A488-Labelled phage was isolated from free dye by passing over two PD10 desalting columns (GE Healthcare).

### Cryo-EM grid preparation

Cryo-EM grids were prepared by pipetting 2.5 μl of sample onto glow-discharged Quantifoil grids (Cu/Rh R2/2, 200 mesh) and plunge frozen

into liquid ethane using a Vitrobot Mark IV (ThermoFisher). For cryo-ET only, 7 μl of sample was mixed with 1 μl of protein-A conjugated with 10 nm colloidal gold (CMC, Utrecht) before 2.5 μl was applied to the grid. Plunge-frozen grids were transferred to and stored in liquid nitrogen until imaging.

### Cryo-EM and cryo-ET data collection

Cryo-EM data for screening specimens were collected using a Talos Arctica (ThermoFisher) operated at 200 kV. High-throughput data was collected on a Titan Krios microscope operated at 300 kV fitted with a Quantum energy filter (slit width 20 eV) and a K3 direct electron director (Gatan) operating in counting mode at an unbinned, calibrated pixel size of 1.1 Å using the EPU software. A combined total dose of approximately 53.9 e⁻/A² per exposure was applied with each exposure lasting 3.6 s and 40 frames were recorded per movie. In total 4259 movies were collected between −1 and −3 μm defocus. Tilt series data for cryo-ET was collected on a Titan Krios using the Quantum energy filter and K2 direct electron director with SerialEM software[51]. Tilt series were collected in two directions starting from 0° between ±60° with a 1° tilt increment, acquired with defoci ranging from −4 to −5 μm, with a combined dose of approximately 120 e⁻/Å² applied over the entire series. Tilt series were collected at an unbinned calibrated pixel size of 4.41 Å. Tomograms were reconstructed using the etomo package[51]. The final tomograms were filtered using the tom_deconv.m script[52]. For measuring filament density in the Pf4 tomogram, a box with 101 pixels cube length (corresponding to a box length of 89.72 nm; box volume 722,217 nm³) was cropped out. Within this box, filaments were manually counted, resulting in 69 counted filaments spanning the box. Based on box and filament geometry, assuming that Pf4 phage filament is a cylinder with the radius 3.1 nm, the filaments were calculated to occupy 25.88% of the box. For fd, filament density was calculated using a 6 nm spacing as indicated in the Fourier transform (Fig. 4). Given an equal spacing between filaments, this results in 225 filaments for a box of an equivalent size to Pf4. Using a filament radius of 3.2 nm, we obtain a volume occupancy by fd of 89.92%.

### Cryo-EM data processing and model building

Helical reconstruction was performed in RELION 3.1[53–56]. Movies were motion-corrected using the RELION implementation of MotionCor2[57], and defocus was estimated using CtfFind4[58]. Using a previously established starting symmetry of fd of 36° and 16 Å[12,14], iterative rounds of 3D refinement and 3D classification were used to create a reference with alpha-helical features that enabled refinement to <5 Å resolution. CTF (contrast transfer function) refinement and Bayesian polishing with CTF-multiplication were then applied to create a 3.2 Å resolution map. Symmetry searches were used during classification and refinement, and a final rotation of 35.46° and rise of 16.62 Å per subunit were obtained. A model of the full mature fd capsid protein was manually built in Coot[59], and real-space refinement against the cryo-EM density was performed in PHENIX[60] using 50 capsid subunits with symmetry restraints (see Supplementary Table 1 for processing statistics). Residues 1–5, which are not well-resolved in the cryo-EM density, are excluded in the final refinement (PDB-8CH5).

### Theoretical scaling calculation of tactoid geometry

We performed a theoretical scaling calculation, involving minimising an approximate free energy *F* of a tactoid that comprises of hard rods (modelled phages)[34]. By assuming that the volume of a tactoid is much greater than the volume of an individual rod, we can decompose the total tactoid free energy into two terms: one term describing the surface (interfacial) contribution and a second term describing the volumetric (elastic) contribution.

In order to arrive at a free energy that is simple enough to solve, yet complex enough to retain information on the overall geometry of the rods and tactoids, we make further physical assumptions. First, we

use what is called the one-constant approximation for the elastic term[61]. This means that the splay and bend elastic effects contribute to the free energy in equal measure. Second, we ignore twist and saddle-splay effects, as we are primarily concerned with large changes in overall shape and size, not on complex internal organisation of the rods. Lastly, we assume that the tactoid is sufficiently elongated (not spherical) and that there is a slight curvature in the stacking of rods in the tactoid, as observed in our experiments. Given the above assumptions, the free energy is given by

$$F \sim \gamma A + \frac{KV}{R^2},\quad (4)$$

where $\gamma$ is the surface tension, $A(r, R)$ is the area of the tactoid, $V(r, R)$ is the tactoid volume, $K$ is the Franck elastic constant, $r$ is the short (minor) axis of the tactoid, and $R$ is the long (major) axis of the tactoid.

To find the equilibrium geometry of the tactoid, we minimise $F$ with respect to both $r$ and $R$ at fixed volume. In doing so we approximate the area as $A \sim Rr$ and the volume as $V \sim Rr^2$. The equations to satisfy are given by:

$$\frac{\partial F(r,R)}{\partial r} - \lambda \frac{\partial V(r,R)}{\partial r} = 0,\quad (5a)$$

$$\frac{\partial F(r,R)}{\partial R} - \lambda \frac{\partial V(r,R)}{\partial R} = 0,\quad (5b)$$

where $\lambda$ is a Lagrange multiplier. By solving the above equations, rearranging them for $\lambda$, and then setting them to be equal, we obtain the following relation:

$$\frac{R}{r} \sim \left(\frac{K}{\gamma}\right)^{3/5} V^{-1/5}.\quad (6)$$

Next, to involve the shapes of the hard rods we approximate the Frank elastic constant and interfacial tension with their values at the homogeneous-bipolar transition given by van der Schoot[62]:

$$K \sim \rho \frac{b}{a^2},\quad (7a)$$

$$\gamma \sim (ab)^{-1},\quad (7b)$$

where $\rho$ is the dimensionless volume fraction (also known as packing fraction) of hard rods in a tactoid. Given this, we reach the final relation given by:

$$\frac{R}{r} \sim \left(\rho \frac{b^2}{a}\right)^{3/5} V^{-1/5}.\quad (8)$$

**Antibiotic protection assay**

An overnight culture of PAO1 *ΔPAO728* was grown in LB media at 37 °C, diluted 1 in 100 into LB medium and grown at 37 °C to an $OD_{600}$ of 0.5. 100 μl of the resulting culture was added to a 96-well plate and grown for a further 30 min at 37 °C. 100 μl of the indicated phage and/or polymer components were added to the culture such that final concentrations of components were: sodium alginate (Scientific Laboratory Supplies) (4 mg/ml), fd (1 mg/ml) and Pf4 (1 mg/ml). Additionally, tobramycin (10 μg/ml) (Sigma) was added as indicated and cultures grown further for 3 h before a 10 μl sample for each assay condition was taken, serially diluted 10-fold and 100 μl of the dilutions plated onto LB agar plates. Plates were incubated overnight at 37 °C and colonies forming units (cfu) enumerated. Experiments were performed in triplicate. Mean cfu/ml with standard deviation were calculated and plotted using Prism GraphPad software.

**Fluorescent recovery after photobleaching (FRAP) experiments on phage tactoids**

FRAP experiments were conducted on fd tactoids (1 mg/ml fd phage, 4 mg/ml alginate) and Pf4 tactoids (1 mg/ml Pf4 phage, 4 mg/ml alginate) incubated for 1 h at RT after mixing. Tactoids were loaded into 0.17 mm thick glass capillaries (Warner Instruments) and imaged on a Zeiss LSM780UV confocal microscope. FRAP was performed using the Zen software bleaching module. Laser power of 80% was used for photobleaching events with a laser power of 2% used for imaging of all steps apart from photobleaching. A custom circular field of view was selected for photobleaching event which occurred after 5 frames. Frames were recorded with a time interval of 1.62 s and a total imaging time of 46 s. Time series were drift corrected using Fiji StackReg plugin using rigid body registration. Fluorescent intensity was measured in Fiji by plotting Z-axis profiles. Fluorescent intensities were normalised to 100%, pre-photobleaching, and 0% after photobleaching. Ten recovery curves were averaged and plotted in Microsoft Excel and Graphpad Prism and curves fitted to $A(1 - e^{-t/T})$ to calculate half-life of recovery[63]. Movie and figure panels were prepared using Fiji.

**Fluorescence microscopy**

**Phage tactoids.** To obtain equivalent levels of total tactoid formation (summed area of all tactoids), A488-labelled fd phage (final concentration 0.5 mg/ml) or A488-labelled Pf4 phage (final concentration 2.7 mg/ml) were mixed with sodium alginate or hyaluronan (final concentration 4 mg/ml) and incubated at room temperature for 24 h. 5 μl of the resulting sample was pipetted onto 0.7% (w/v) agar pads constructed using 15 × 16 mm Gene Frames (ThermoFisher) following the manufacturer's protocol, with a coverslip placed on top. The slide was imaged using a Zeiss Axioimager M2 (Carl Zeiss) microscope in both brightfield and fluorescence mode. Quantification of individual tactoid area and morphology was performed using MicrobeJ[64]. Maxima corresponding to individual tactoids were analysed using MicrobeJ shape analysis functions to quantify tactoid area, major axis length and minor axis length. Experiments were performed in triplicate. Presented images were background subtracted and figure panels prepared using Fiji. Graphs were plotted using Prism GraphPad software.

**Phage tactoids/bacterial cells.** *P. aeruginosa* PAO1 *ΔPAO728* or *E. coli* XL1, a K12 derivative, were grown to an $OD_{600}$ of 0.5 and incubated with A488-labelled phage (final concentration 1 mg/ml) and sodium alginate (final concentration 4 mg/ml) in PBS (137 mM NaCl) for 3 h. Five μl of the sample was pipetted onto 1.5% (w/v) agar pads constructed using 15 × 16 mm Gene Frames (ThermoFisher) with a coverslip applied, and imaged using a Zeiss Axioimager M2 microscope (Carl Zeiss). Images were background subtracted and figure panels prepared using Fiji.

**Texas Red-gentamicin (GTTR) diffusion into bacterial cells**

PAO1 *ΔPAO728* or *E. coli* XL1 were grown to an $OD_{600}$ of 0.5 and incubated with A488-labelled phage (final concentration 1 mg/ml) and sodium alginate (final concentration 4 mg/ml) for 30 min. Texas Red-gentamicin (AAT Bioquest) was added to a final concentration of 1 μM and incubated for a further 4 h. 5 μl of the sample was pipetted onto 1.5% (w/v) agar pads constructed using 15 × 16 mm Gene Frames (ThermoFisher) with a coverslip applied, and imaged using a Zeiss Axioimager M2 microscope (Carl Zeiss). Images were background subtracted and figure panels prepared using Fiji. Images were quantified by semi-automated segmentation of bacterial cells and associated tactoid (see below) and fluorescent intensity in the Texas Red channel was measured at the coordinates of segmented bacteria.

## Semi-automated segmentation of images with phage tactoids surrounding bacterial cells

Bacteria were manually selected from brightfield channel images to identify the positions of cells. Bacterial cell shapes were found using the *activecontour* algorithm in MATLAB[65]. The regions of identified bacteria were dilated and used as seed inputs for the segmentation of the tactoids in the fluorescence channel using the *activecontour* algorithm. The segmented bacterial cells from the bright field and the tactoids from the fluorescence channel were then used to calculate morphological parameters.

## Preparation of molecular dynamics simulation systems

The previously published Pf4 capsid cryo-EM structure (PDB: 6TUQ)[4] and the fd capsid cryo-EM structure described in this paper were cropped using VMD[66] and PyMOL[67] to prepare simulations of a smaller and computationally judicious system. The Pf4 and fd capsid systems were centred in a $12.7\,nm^3$ and $12.6\,nm^3$ box respectively. Both systems were solvated using the TIP3P water model[68] and neutralised in 0.15 M NaCl. Further information on simulation set up can be found in Supplemental Table 2. One round of steepest descent energy minimisation was performed on each system for 100 ps, followed by one 5 ns NVT and one 5 ns NPT equilibration, with restraints applied to the heavy backbone atoms.

## Atomistic molecular dynamics simulations

Simulations were performed using the Gromacs 2021.3 simulation package[69] (https://doi.org/10.5281/zenodo.5053220) and the CHARMM-36 forcefield[70]. The CHARMM-36 forcefield was chosen as this is an appropriate forcefield for ion-binding analysis in a simulation system of this size[70]. Atomistic simulations were run in quadruplicate for 250 ns using different starting velocities for each repeat production run and a 2 fs timestep. Periodic boundary conditions were applied and the velocity-rescale thermostat[71], with a coupling constant of 0.2 ps, was used to maintain temperature at 310 K. Pressure was maintained at 1 bar using the Parrinello-Rahman barostat[72] with a coupling constant of $T_P = 2.0$ ps and compressibility of $4.5 \times 10^{-5}\,bar^{-1}$. Long range electrostatic interactions were modelled using the Particle-Mesh Ewald (PME) method[73] and a 1.2 nm cut-off was used. Van der Waals interactions were switched between 1.0 and 1.2 nm using the force-shift modifier. Dispersion correction was not applied. The LINCS algorithm[74] was used to constrain bonds to their equilibrium values. During both Pf4 and fd capsid simulation production runs, 50 kJ/mol/$nm^2$ restraints were applied in xyz on the bottom and top c-alpha carbons of the protein ($z < 3$ or $z > 9$). See Supplementary Fig. 5a for views of restrained atoms, and Supplementary Fig. 5b, c for simulation RMSD values and total values of ion binding over the simulated time, which highlights system convergence. To evaluate total ion binding over time, ions within a 0.5 nm cut off of the protein were selected using gmx select tool[69] (https://doi.org/10.5281/zenodo.5053220). Further analysis was performed using a PyLipID script[75], with modified cut-offs (0.3,0.5) for ion binding, gmx tools[69] (https://doi.org/10.5281/zenodo.5053220) and VMD volmap[66]. For all volmap and PyLipID analyses, trajectories sampled every five frames from each repeat were used for the analysis, totalling 20,000 frames per capsid simulation system. For PyLipID analysis, the trajectories were further sampled by 10 frames for Pf4 and 40 frames for fd. For quantification of ions within the lumen, the cylayer command within MDAnalysis[76,77] was used to count ions within cylindric layers of 1 Å intervals positioned on the centre of geometry of the capsid atoms. Analysis was conducted every 5 ns from the full 250 ns trajectories to a maximum distance of 60 Å. Graphs were plotted using Plotly (Plotly Inc.) and Prism GraphPad software. Preparation of systems for simulation was performed using computing resources at The Kavli Institute in the Structural Bioinformatics and Computational Biochemistry unit at the University of Oxford. Production runs were simulated for 250 ns using computing resources on ARCHER2.

## Statistical analysis

Statistical analysis was performed using Prism GraphPad software and an unpaired *t* test was used to calculate *p*-values.

## Reporting summary

Further information on research design is available in the Nature Portfolio Reporting Summary linked to this article.

## Data availability

The cryo-EM density map of the fd phage capsid generated in this study has been deposited in the Electron Microscopy Databank (EMDB) under the accession number EMD-16657. The corresponding atomic coordinates are deposited in the Protein Data Bank (PDB) under accession code 8CH5. The atomic coordinates of other phage capsids used for comparison are available on the PDB under accession codes 6TUP (cryo-EM structure of Pf4), 6A7F (cryo-EM structure of IKe phage), 2HI5 (previous fd phage cryo-EM model), 2C0X (fibre diffraction/ssNMR model of fd), 1IFI (fibre diffraction model of fd), 1NH4 (ssNMR model of fd). MD data is available on Zenodo (https://doi.org/10.5281/zenodo.10175088). Source data are provided with this publication. All other data are available from the corresponding authors upon request.

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

## Acknowledgements

T.A.M.B. would like to thank UKRI MRC (Programme MC_UP_1201/31), the Human Frontier Science Programme (Grant RGY0074/2021), the EPSRC (Grant EP/V026623/1), the Vallee Research Foundation, the European Molecular Biology Organisation, the Leverhulme Trust and the Lister Institute for Preventative Medicine for support. P.P. is supported by a UKRI Future Leaders Fellowship [MR/V022385/1]. This project made use of time on ARCHER2 granted via the UK High-End Computing Consortium for Biomolecular Simulation, HECBioSim (www.hecbiosim.ac.uk), supported by EPSRC (grant no. EP/R029407/1). We would like to thank Professor Mark Sansom and The Kavli Institute Structural Bioinformatics and Computational Biochemistry unit at the University of Oxford for facilitating the Molecular Dynamics performed in this research. We thank Jeanne Salje and Paul Edelstein for critical comments and the MRC-LMB cryo-EM and light microscopy facility for technical support.

## Author contributions

Conceptualisation: J.B., A.K.T., T.A.M.B.; Methodology: J.B., L.K.D., P.P., A.K.T., T.A.M.B.; Formal Analysis: J.B., M.G., S.C.L., L.K.D., U.S., R.A.C., P.J.S., P.P., A.K.T., T.A.M.B.; Investigation: J.B., M.G., S.C.L., L.K.D., P.P., A.K.T., T.A.M.B.; Writing – Original Draft: J.B., A.K.T., T.A.M.B.; Writing – Review & Editing: J.B., M.G., S.C.L., L.K.D., U.S., R.A.C., P.J.S., P.P., A.K.T., T.A.M.B.; Supervision: R.A.C., P.J.S., P.P., A.K.T., T.A.M.B.; Funding Acquisition: P.J.S., P.P., T.A.M.B.

## Competing interests

The authors declare no competing interests.
