## [Peer review file · Nature Communications]

REVIEWER COMMENTS

Reviewer #1 (Remarks to the Author):

This is a very interesting study that aims to characterize the formation of meso-scale assemblies by different phages and their potential symbiotic role in protecting bacteria against antibiotics. It seems to be an excellent fit for a journal with broad readership. The experiments and quantitative analysis are thorough and well executed. I mainly have specific comments that may help to improve clarity for a general audience.

However, and as acknowledged by the authors in their discussion, something is missing to arrive at new understanding of the mechanisms of antibiotic protection. Perhaps the authors' statement in the abstract "factors governing the formation of such droplets and the mechanism of antibiotic protection are poorly understood" and the introduction build an expectation for this to be answered. The authors provide a new structure of fd capsid and use this to beautifully show that the formation of phage droplets (of different geometries related to the phage geometry) protect any bacterium embedded or encapsulated in them from antibiotic stress. The authors have described this phase droplet protection phenomenon in a previous publication. Here, in their discussion they speculate that the viscous surrounding created by droplets leads to reduced antibiotic diffusion and therefore access to the bacterial cell. Would this not be a testable experiment? Can the authors not design an experiment with a dye of similar chemistry to the drug or a fluorescent antibiotic and measure penetration into the droplet/bacterium? If this is not possible, I would strongly recommend that the authors clearly state from the beginning that this remains an answered question (basically toning down the statement "This study... provides insights into how filamentous molecules protect bacteria from extraneous molecules under crowding conditions).

Specific comments:

1. Line 84: "A unique property of inoviral phages including fd and Pf4 is that they can self-assemble to form ordered but dynamic liquid crystalline droplets under crowding conditions outside the cell." It would be beneficial to introduce how is the crystalline phase (experimentally) defined, what interactions or forces are known to drive such assemblies, and what is the concrete evidence for a the liquid phase (authors refer to droplets throughout the manuscript).
2. Line 93 "Pf4 was subsequently found to form liquid crystalline droplets in the biopolymer-rich extracellular matrix of biofilms" it is not entirely clear whether presence of the biofilm molecules is required for droplet formation.
3. Related to the above 2 point, while the authors have done systematic comparisons between the structures and biochemical properties of 2 phages, it remains an open question throughout the manuscript, and would be useful to describe what are the known characteristics of molecules that drive assembly of phages into droplets.
4. I find the description in line 143 to be confusing, since the author do show the DNA density at lower threshold: "Clear density for the DNA was not resolved in our fd cryo-EM map (Figure S1f),... where an unfeatured central DNA density was reported." Please rephrase for clarity.
5. Line 157 "negatively charged residues are mostly located in the disordered N-terminus (sequence AEGDD), while the negative, outer residues in Pf4 are ordered (Figure 2c). Do these residues contribute to the assembly into meso-scale droplets? What happens on mutation?"
6. The goal of the MD simulations "To probe the structure further" (line 167) is unclear, especially in the absence of DNA. I am not sure how the authors reach the following conclusion based on the simulations, rather than just an based on interpretation of the structure: "Since our simulations did

not contain ssDNA, these increased negatively charged ions in the fd lumen support our expectation from our cryo-EM structures that DNA arrangement in the class I inoviral phage fd (containing circular ssDNA) is different to that in the class II inoviral phage Pf4 (containing linear ssDNA).” A better description early on in this section of the goal of the experiment would be beneficial.

7. Line 187 “We assembled liquid crystalline droplets by mixing phage with alginate, a biopolymer commonly found in the extracellular matrix of biofilms” Could the authors describe the time scale of assembly . Also some description on the biopolymers (chemical) properties would be beneficial, along the lines of my previous comments. Have the authors tested other biopolymers? Can these change the geometry of the droplets?

8. Line 188 “compared both fd and Pf4 droplets at phage and biopolymer concentrations that gave equivalent levels of total liquid crystalline droplet formation (Figures S5a-c).” is confusing since the authors clearly show different areas occupied by droplets in Figure 4c.

9. “Biochemical phage interactions could affect the physical prefactors in Eq. (1), and thus could enter Eq. (2), but do not seem to underlie the qualitative differences in our measurements of droplet aspect ratios.” Some measure of the fit of experimental vs. theoretical measurements should be provided in Figure 4f.

10. The tomographic data show that fd pack more tightly than Pf4, to a level that even the individual phage boundaries are difficult to discern for fd. Is it possible that such molecular packing differences, that may arise due to slight biochemical difference on the phage surface are some of these “qualitative differences” that contributor for the different droplet morphology? This links to my early questions on molecular interactions that drive droplet formation.

11. Also, do the molecular exchange dynamics differ (measurable by FRAP for example)?

12. This theoretical calculation and the prediction that it is only the geometry of the phage that dictates the droplet geometry also then lends itself to testing this prediction. Can the authors envision a experiments with a different/engineered phage with different/similar properties to substantiate this?

13. Line 254 “suggesting that bacterial cell association is a biophysical process rather than being driven by specific biochemical interactions, since the surfaces of the phages as well as the outer surfaces of *E. coli* and *P. aeruginosa* are biochemically distinct.” It is unclear whether there was any difference in the numbers of bacterial cells associated with the different phage droplets? Fraction of bacteria associated should be provided.

14. Line 258 “Curiously, although both phage droplets could associate with bacterial cells, Pf4 droplets more readily encapsulated both bacteria and were able to completely surround the cell, whereas fd droplets interacted with cells laterally (Figures 5a-f).” Could this be explained by an effect of the different surface tension of the droplets? Please discuss.

15. Line 285 “These antibiotic protection assays suggest that biophysical parameters govern and profoundly affect bacterial responses to external stresses such as antibiotic treatment.” Does this suggest that encapsulation by any biomaterial (phage or else) would result in protection? Please clarify.

16. Line 293 “Numerous inconsistent models for the fd phage capsid have been proposed over the years by X-ray fibre diffraction, ssNMR and -resolution cryo-EM”. It would be beneficial that the authors explain what may be the source of the differences. Is it related to different constructs/reconstitutions? Please discuss.

17. Line 328 “Pf4 droplets almost fully encapsulate rod-shaped *P. aeruginosa* and *E. coli* cells,

compared to the artificial case of fd droplets, where only lateral interactions were seen." Why is the later defined artificial? Please clarify.

18. Line 336 "One of the most striking results in this study was that droplet association with cells is sufficient for antibiotic protection (Figure 6), rather than complete encapsulation." It would be useful to visualize this at higher resolution (tomography?) to be able to make this conclusion. It is possible that a small amount of phages are sufficient to block diffusion of the antibiotic.

19. Figure 3: the standard error of the mean (SEM) plotted in transparent colour is impossible to see, so not clear if it is too small or not included in the plot.

20. Figure S1: I assume arrows in (a) is to indicate individual phages/capsids. Should be clarified in the legend. The variable local resolution in (e) is interpreted in line 711 "Slight differences in local resolution between individual capsid proteins indicate minor structural flexibility within the capsid.", but remain unclear to me how the individual subunits, rather than different regions along each subunit could exhibit different local resolution if the map is generated with symmetrisation.

Reviewer #2 (Remarks to the Author):

In the Ms. entitled "Biophysical basis of phage liquid crystalline droplet-mediated antibiotic tolerance in pathogenic bacteria" the Authors claim to describe biophysical and biochemical parameters that can explain the liquid crystalline droplet mediated antibiotic mechanisms of two inovirus, namely: fd and Pf4. The Authors present a concise summary about what is the actual state of such a research avenue. There are several key points of this investigation, however, many of them are not well interconnected to each other. For example, when the Authors claim about biophysical basis, is this related to any biophysical observable in particular? In the current Ms., I cannot distinguish if such a "biophysical basis" is driven by electrostatics, mechanics, viscosity or any others. Any important outcomes and conclusions in this respect should be highlighted. Moreover, what is the role of Inoviruses genome in the morphological changes between fd and Pf4? In particular when fd and Pf4 are compared?

I recommend addressing the already described major points and the listed ones below (1-9) before further rounds for publication:

1. Equations 1 and 2 require a schematic representation of the very coarse theoretical model for the phages droplets.

2. Section: Atomic structure line 121..: What is the Net charge of the protein for fd and Pf4? What about the surface charges, outside and inside the proteinaceous capsids? Which is the expected charge of the genome? What is the role of the outer surface charge and of the genome in the morphological changes of the filaments arrangements?

3. Section: Droplet encapsulation, line 254..: What is the reasoning behind the phrase: "suggesting that bacterial cell association is a biophysical process rather than being driven by specific biochemical interactions, since the surfaces of the phages as well as the outer surfaces of E. coli and P. aeruginosa are biochemically distinct." Chemically distinct atoms could be attractive exactly if they are different, or? What type of biophysical process is the association of Bacterial cells and the lc droplets? The proposed theory is only macroscopical (highly hypothetical) and hence of course not considering any chemical information. This point need to be clarified

4. Regarding the experimental setup for using E.coli and P.aeruginosa, which are the details in terms

of salt concentrations in each system mM? how are the apparent higher charges in fd affecting the adherence of the E.coli and P.aeruginosa. Which are the main differences of both systems (E.coli and P.aeruginosa)? they need to be also depicted at least schematically in e.g. Figure 7.

5. The authors need to perform contact angle measurements of the crystalline droplets, to support their analysis on the droplet formation.

6. I strongly recommend the authors to perform a genome structural analysis to verify the linear and circular ssDNA assumptions, this can be rapidly carried out by using a similar method as the one described here (<https://doi.org/10.3390/v13081555>); where the effects of the charges (lines 305-317) can be theoretically confirmed.

7. Another important missing point in each of the results sections are the limitations of the analysis, for example the very coarse theoretical model, or if any symmetry is imposed in the cryoEM structures, among many others related to the droplet and experimental observations.

8. Conclusions and further research questions are missing.

9. Table S1: data collection and processing statistics for fd capsid structure
—> data collection and processing statistics for Pf4 missing!!!

Response to reviewers for Böhning *et al*/ NCOMMS-23-03978-T

Green: Response Text

Blue: Manuscript excerpt

Reviewer #1 (Remarks to the Author):

This is a very interesting study that aims to characterize the formation of meso-scale assemblies by different phages and their potential symbiotic role in protecting bacteria against antibiotics. It seems to be an excellent fit for a journal with broad readership. The experiments and quantitative analysis are thorough and well executed. I mainly have specific comments that may help to improve clarity for a general audience.

However, and as acknowledged by the authors in their discussion, something is missing to arrive at new understanding of the mechanisms of antibiotic protection. Perhaps the authors' statement in the abstract "factors governing the formation of such droplets and the mechanism of antibiotic protection are poorly understood" and the introduction build an expectation for this to be answered.

We thank the reviewer for their helpful feedback which has greatly strengthened the manuscript. We have performed experiments showing reduced antibiotic diffusion in the presence of phage liquid crystalline droplets in the revised version of the article, providing mechanistic insight into the mode of antibiotic protection. We have also adjusted the abstract and introduction to better reflect the results reported in the manuscript.

Summary, L42: "We show that fd and Pf4 form liquid crystalline droplets with different morphologies, governed by emergent droplet material properties, which are determined by phage geometry and packing density. Finally, we show that droplets formed by either phage protect rod-shaped bacteria from antibiotic treatment, and that direct association with a droplet is required for protection, suggesting formation of a diffusion barrier by the droplet. This study advances our understanding of phage assembly into liquid crystalline droplets, providing multi-scale insights into how filamentous molecules protect bacteria from extraneous molecules under crowding conditions found in biofilms or on infected host tissues."

L112: "We further show that fd and Pf4 form liquid crystalline droplets with different morphologies and develop a theoretical framework, which shows that morphological

differences between droplets are governed by differences in droplet material properties, which are in turn determined by phage geometry and packing. We show that droplets of both fd and Pf4 associate with bacterial cells, conferring protection against antibiotic treatment, suggesting the formation of a diffusion barrier. Our results explain the factors determining liquid crystalline droplet properties, with implications for antibiotic protection provided by filamentous molecules, which are enriched in viscous environments such as bacterial biofilms or sites of infection in hosts.”

The authors provide a new structure of fd capsid and use this to beautifully show that the formation of phage droplets (of different geometries related to the phage geometry) protect any bacterium embedded or encapsulated in them from antibiotic stress. The authors have described this phase droplet protection phenomenon in a previous publication. Here, in their discussion they speculate that the viscous surrounding created by droplets leads to reduced antibiotic diffusion and therefore access to the bacterial cell. Would this not be a testable experiment? Can the authors not design an experiment with a dye of similar chemistry to the drug or a fluorescent antibiotic and measure penetration into the droplet/bacterium? If this is not possible, I would strongly recommend that the authors clearly state from the beginning that this remains an answered question (basically toning down the statement “This study... provides insights into how filamentous molecules protect bacteria from extraneous molecules under crowding conditions).

As suggested, we have measured diffusion of a fluorescently labelled antibiotic (Texas Red-labelled gentamicin) into cells in the presence of phage liquid crystalline droplets. Our new results show that cells encapsulated by droplets show significantly lower uptake of antibiotics, compared with bacteria in the same sample that are not encapsulated (revised Figure 6 and new Figure S9). This suggests that droplets may form a diffusion barrier around cells, resulting in protection against antibiotics.

“Figure 6: Association with liquid crystalline droplets protects bacterial cells from antibiotic uptake, suggesting the formation of a diffusion barrier.

[...] **c-h)** Fluorescence and light microscopy images of Alexa 488-labelled Pf4/fd phage liquid crystalline droplets incubated with Texas Red-labelled gentamicin (GTR) for 4 hours. Shown are Alexa488 phage signal (Cyan – fd, Magenta – Pf4) and pseudocoloured cells (yellow) as determined through brightfield light microscopy (left), and Texas Red signal corresponding to uptake of the fluorescently labelled antibiotic GTR by cells (right). Numbering indicates site of cells in corresponding images. **i)** Plot quantifying GTR uptake after 4 hours respectively in conditions indicated on the x-axis. Bar shows the mean of three independent experiments and error bars represent standard deviation. Association with both Pf4 and fd liquid crystalline droplets results in significantly reduced antibiotic uptake as compared to a control with no phage (fd $P_{\text{value}} < 0.0001$, Pf4 $P_{\text{value}} < 0.0001$), and as compared to cells in the same sample that are not droplet-associated (fd $P_{\text{value}} < 0.0001$, Pf4 $P_{\text{value}} < 0.0001$).”

Specific comments:

1. Line 84: “A unique property of inoviral phages including fd and Pf4 is that they can self-assemble to form ordered but dynamic liquid crystalline droplets under crowding conditions outside the cell.” It would be beneficial to introduce how is the crystalline phase (experimentally) defined, what interactions or forces are known to drive such assemblies, and

what is the concrete evidence for a the liquid phase (authors refer to droplets throughout the manuscript).

We now elaborate on the characteristics of liquid crystalline droplets and the drivers for their formation for Pf4 and fd in the revised manuscript. We have also added further citations to seminal past studies on this subject that define the phase behaviour of colloidal rods. We have also performed new FRAP experiments showing dynamic behaviour of the phages inside the droplets (discussed below in another point).

L 81: “A unique property of inoviral phages, including fd and Pf4, is that they can spontaneously and reversibly self-assemble to form ordered but dynamic liquid crystalline droplets under crowding conditions¹⁻³. Within droplets, laterally associated phages are orientationally aligned, but not regularly ordered as in a crystal². Phages within the liquid crystalline droplets are mobile, as shown by fluorescence recovery after photobleaching (FRAP) experiments on Pf4 droplets⁴. Assembly of phages into droplets is driven by a depletion interaction, an effective attractive force between rigid constituents arising from the preferential exclusion of solvent elements (depletants) from the vicinity of the larger phage constituents^{3,5}. In the presence of high-molecular weight crowding polymers as a depletant and counterions to compensate surface charges, phages spontaneously assemble into liquid-crystalline droplets. As expected from a depletion interaction, the size of the crowding polymer positively correlates with the extent of liquid crystalline droplet formation¹. This process is thermodynamically favourable as phage alignment allows for higher degrees of freedom of the crowding polymer, increasing the total entropy^{3,6}.”

2. Line 93 “Pf4 was subsequently found to form liquid crystalline droplets in the biopolymer-rich extracellular matrix of biofilms” it is not entirely clear whether presence of the biofilm molecules is required for droplet formation.

We have now explicitly stated that the presence of crowding agents is essential for the formation of liquid crystalline droplets both *in vitro* and in biofilms.

L89: “In the presence of high-molecular weight crowding polymers as a depletant and counterions to compensate surface charges, phages spontaneously assemble into liquid-crystalline droplets. As expected from a depletion interaction, the size of the crowding polymer positively correlates with the extent of liquid crystalline droplet formation¹. This process is thermodynamically favourable as phage alignment allows for higher degrees of freedom of the crowding polymer, increasing the total entropy^{3,6}.”

3. Related to the above 2 point, while the authors have done systematic comparisons between the structures and biochemical properties of 2 phages, it remains an open question throughout the manuscript, and would be useful to describe what are the known characteristics of molecules that drive assembly of phages into droplets.

Apologies for the confusion. We refer to our answers to points 1 and 2 above. As requested, we have added more introductory text in the revised manuscript to make clear to the readers what is known from previous studies.

4. I find the description in line 143 to be confusing, since the author do show the DNA density at lower threshold: “Clear density for the DNA was not resolved in our fd cryo-EM map (Figure S1f),... where an unfeatured central DNA density was reported.” Please rephrase for clarity.

While we see density corresponding to DNA, it is smeared out and does not allow model building of the DNA itself, meaning we cannot resolve contacts between the phage capsid and DNA, also the case in previous studies on Class I inoviruses⁷. We have clarified this in the revised manuscript:

L146: “While density for the DNA is observed in our fd cryo-EM map, it is not well-resolved and does not support atomic model building of the DNA genome (Figure S1f). Poorly resolved DNA density was also observed in previous studies on class I inoviral bacteriophages (Figure S2), where direct atomic model building into the DNA density could also not be performed⁷.”

5. Line 157 “negatively charged residues are mostly located in the disordered N-terminus (sequence AEGDD), while the negative, outer residues in Pf4 are ordered (Figure 2c). Do these residues contribute to the assembly into meso-scale droplets? What happens on mutation?”

Depletion attraction occurs between similarly charged objects, such as rod-shaped inoviruses, as long as sufficient counter-ions are present in the solution^{1,2}, which we now clarify in the introduction (see points 1 and 2 above).

L89: “In the presence of high-molecular weight crowding polymers as a depletant and counterions to compensate surface charges, phages spontaneously assemble into liquid-crystalline droplets.”

We do not expect any major effect upon mutation of these residues on the assembly of droplets. While the use of two different phages has allowed us to probe the effect of biochemical differences, genomic engineering of the phage would be challenging because phage capsid residues have to fulfil several other roles including viral assembly and interactions with co-factors. Genomic engineering is thus not straight-forward and is beyond the scope of this study. We now discuss the possibility of these experiments in the discussion.

L408: “Furthermore, mutation of the surface charges could enlighten how far significant alteration of the phage surface would perturb the biophysical effects described in this study.”

6. The goal of the MD simulations “To probe the structure further” (line 167) is unclear, especially in the absence of DNA. I am not sure how the authors reach the following conclusion based on the simulations, rather than just an based on interpretation of the structure: “Since our simulations did not contain ssDNA, these increased negatively charged ions in the fd lumen support our expectation from our cryo-EM structures that DNA arrangement in the class I inoviral phage fd (containing circular ssDNA) is different to that in the class II inoviral phage Pf4 (containing linear ssDNA).” A better description early on in this section of the goal of the experiment would be beneficial.

We thank the reviewer for this comment and now clarify that the MD is meant to be complementary and supportive to the insights gained from the structure itself.

L185: “To complement our structural studies, we performed atomistic molecular dynamics (MD) simulations of fd and Pf4 phage capsids to determine whether the additional positively charged residues in the fd lumen can interact with higher amounts of negative charges than the Pf4 capsid.”

7. Line 187 “We assembled liquid crystalline droplets by mixing phage with alginate, a biopolymer commonly found in the extracellular matrix of biofilms” Could the authors describe the time scale of assembly . Also some description on the biopolymers (chemical) properties would be beneficial, along the lines of my previous comments. Have the authors tested other biopolymers? Can these change the geometry of the droplets?

The assembly of liquid crystalline droplets through depletion attraction is essentially instantaneous and depends on the concentrations of phage and biopolymer used, explored previously by us and others^{1,4}.

L81: “A unique property of inoviral phages, including fd and Pf4, is that they can spontaneously and reversibly self-assemble to form ordered but dynamic liquid crystalline droplets under crowding conditions¹⁻³.”

L89: “In the presence of high-molecular weight crowding polymers as a depletant and counterions to compensate surface charges, phages spontaneously assemble into liquid-crystalline droplets.”

As requested, we now provide data of fd and Pf4 droplet formation in the presence of a different crowding agent, the common host-biopolymer hyaluronan, showing equivalent morphology of the respective phage droplets for both biopolymers (new Figure S6).

“Figure S6: Liquid crystalline droplet formation of Pf4 and fd in the presence of alginate versus hyaluronan.

a-b, d-e) Light microscopy images of fluorescently-labelled fd (cyan) or Pf4 (magenta) liquid crystalline droplets formed in the presence of either alginate or hyaluronan as crowding biopolymer. c,f) Violin plot of measured aspect ratios of liquid crystalline droplets formed in alginate versus hyaluronan. Dotted lines indicate mean and 25th/75th percentiles.”

8. Line 188 “compared both fd and Pf4 droplets at phage and biopolymer concentrations that

gave equivalent levels of total liquid crystalline droplet formation (Figures S5a-c).” is confusing since the authors clearly show different areas occupied by droplets in Figure 4c.

We apologise for the confusion. We used concentrations of phages and biopolymers that produced the same overall area of droplets for fd and Pf4 (Figure S5a-c), which is necessary to compare individual droplet morphology under equivalent conditions. Hence, while the morphologies of individual droplets are different, the overall droplet area, meaning the sum total of the area of all droplets in a given field of view of the specimen, is the same (within experimental error). We now clarify this in the figure by changing the Y-axis labelling, and in the figure legend (Figure 4c).

“**[Figure 4] c)** Bar chart showing the average area of individual liquid crystalline droplets as assessed by light microscopy followed by segmentation of liquid crystalline droplets. Values shown are the mean of three independent experiments and error bars represent standard deviation.”

9. “Biochemical phage interactions could affect the physical prefactors in Eq. (1), and thus could enter Eq. (2), but do not seem to underlie the qualitative differences in our measurements of droplet aspect ratios.” Some measure of the fit of experimental vs. theoretical measurements should be provided in Figure 4f.

We have added standard deviations for the fitting parameters in the caption of Figure 4g (previously Figure 4f) and mention them in the main text:

“**[Figure 4] g)** Liquid crystalline droplet aspect ratio as a function of droplet volume. Droplets, as visualised via light microscopy, follow the scaling law $R/r \propto V^{-1/5}$ as shown through lines of best fit to $R/r = CV^{-1/5}$, where C is the fitting parameter ($C_{Pf4} = 5.88 \pm 0.11 \mu m^{3/5}$ and $C_{fd} = 3.72 \pm 0.09 \mu m^{3/5}$).”

L242: “As predicted, through lines of best fit to $R/r = CV^{-1/5}$, where C is the fitting parameter ($C_{Pf4} = 5.88 \pm 0.11 \mu\text{m}^{3/5}$ and $C_{fd} = 3.72 \pm 0.09 \mu\text{m}^{3/5}$, we observed that the aspect ratios and volumes of both the fd and Pf4 droplets displayed this relationship (Figure 4f-g).”

10. The tomographic data show that fd pack more tightly than Pf4, to a level that even the individual phage boundaries are difficult to discern for fd. Is it possible that such molecular packing differences, that may arise due to slight biochemical difference on the phage surface are some of these “qualitative differences” that contributor for the different droplet morphology? This links to my early questions on molecular interactions that drive droplet formation.

We have amended the text to clarify how biochemical interactions enter our scaling theory:

L262: “The small quantitative discrepancy between the predicted and measured values of C_{Pf4}/C_{fd} could be caused by differences in biochemical phage-phage interactions (e.g., electrostatic surface interactions; see Figure 2) or through distinct phage geometry causing differing depletion interactions. While these interactions already feed into Eq. (2) through the phage packing fraction ρ , such interactions also affect the material properties K and γ in Eq. (1), and would therefore enter Eq. (2) as prefactors. However, the reasonably good agreement between experiment and theory suggests that differences in K and γ between Pf4 and fd tactoids are mostly determined by phage aspect ratio.”

11. Also, do the molecular exchange dynamics differ (measurable by FRAP for example)?

We have now performed FRAP experiments on both Fd and Pf4 liquid crystals and see similar exchange dynamics in the experiment (new Figure S7 and new Movie S3).

“Figure S7: Fluorescence recovery after photobleaching (FRAP) for Pf4 and fd liquid crystalline droplets.

a-f) Fluorescence images of liquid crystalline droplets before (left), immediately after (middle), and 6s after photobleaching for fd (**a-c**) and Pf4 (**d-f**) liquid crystalline droplets. **g)** Fluorescence recovery curves for fd and Pf4, normalised to fluorescence at 0 s. The resulting half-life of recovery is shown.”

12. This theoretical calculation and the prediction that it is only the geometry of the phage that dictates the droplet geometry also then lends itself to testing this prediction. Can the authors envision a experiments with a different/engineered phage with different/similar properties to substantiate this?

We think this is an excellent point for future experiments. Given the wide range of experiments needed for the exhaustive comparison we performed on the structure of the phages, the formation of droplets, their interactions with cells, and protection from antibiotic effect, it was beyond the possibilities of this study to include further phages. Genomically engineering phages is difficult, as any changes in the genome may disrupt phage entry, phage assembly or egress, but should be considered for future lines of enquiry. In the revised manuscript, we discuss the possibility of such experiments in the future.

L406: “Future studies could employ genomically engineered phages with various lengths to further probe the role of phage material properties on liquid crystalline droplet formation and encapsulation. Furthermore, mutation of the surface charges could enlighten how far significant alteration of the phage surface would perturb the biophysical effects described in this study.”

13. Line 254 “suggesting that bacterial cell association is a biophysical process rather than

being driven by specific biochemical interactions, since the surfaces of the phages as well as the outer surfaces of *E. coli* and *P. aeruginosa* are biochemically distinct.” It is unclear whether there was any difference in the numbers of bacterial cells associated with the different phage droplets? Fraction of bacteria associated should be provided.

We now provide the fractions of liquid crystalline droplet-associated bacteria in Figure 5; the difference in association was not statistically significant.

“[Figure 5] d) Percentage of *P. aeruginosa* cells associated with liquid crystalline droplets. Differences in association are not statistically significant. h) Percentage of *E. coli* cells associated with liquid crystalline droplets. Differences in association are not statistically significant.”

14. Line 258 “Curiously, although both phage droplets could associate with bacterial cells, Pf4 droplets more readily encapsulated both bacteria and were able to completely surround the cell, whereas fd droplets interacted with cells laterally (Figures 5a-f).” Could this be explained by an effect of the different surface tension of the droplets? Please discuss.

We agree with the reviewer’s suggestion that it is physically plausible that differences in surface tension could help explain why Pf4 droplets more readily encapsulate bacteria than fd droplets, because of the effect of surface tension on wetting via the Young-Laplace equation. We have amended the text to discuss this in detail (starting from line 283). We thank the reviewer for raising this interesting point.

L308: “We wondered whether the observed qualitative differences in encapsulation of bacterial cells – that Pf4 droplets more readily encapsulated the bacteria as compared to fd droplets (Figure S8f) – could be explained by biophysical phage and droplet properties within our coarse-grained framework. Based on our approximations of the bulk elastic constant K and surface tension γ in terms of the phage packing fraction ρ and phage shape (width a and length b), we predict that Pf4 droplets have a ~ 1.2 times larger K but ~ 4 times smaller surface tension γ than fd droplets. By considering the bacteria as the wetting surface for the phages, the Young-Laplace equation for the interior contact angle θ (ref⁸) gives

$$\cos(\theta) = \frac{\gamma_{b-d} - \gamma_{b-a}}{\gamma}, \quad (3)$$

where γ_{b-d} and γ_{b-a} are the respective bacteria-droplet (assumed constant) and bacteria-alginate (assumed constant) surface tensions, and γ is the alginate-droplet surface tension already defined above. Eq. (3) suggests that a higher alginate-droplet surface tension γ results in an increase in contact angle between the droplet and bacteria interfaces, i.e. reduced wetting. Therefore, we predict reduced wetting (and therefore reduced encapsulation) of bacteria by fd droplets because of their higher surface tension than Pf4 droplets. This is in line with experiments (Figure 5), further confirming that material properties, which are determined by phage shape and packing, drive the observed differences in behaviour between Pf4 and fd droplets.”

15. Line 285 “These antibiotic protection assays suggest that biophysical parameters govern and profoundly affect bacterial responses to external stresses such as antibiotic treatment.” Does this suggest that encapsulation by any biomaterial (phage or else) would result in protection? Please clarify.

We indeed believe that encapsulation by any filaments (such as those present in the extracellular matrix of biofilms) could result in protection and discuss this in the revised manuscript:

L433: “Future research including more biochemically reconstituted components of the biofilm extracellular matrix will be needed to unambiguously prove this proposal and to delineate the exact contribution of each component in biofilm formation. Since filamentous molecules are abundant in most bacterial biofilms⁹, the biophysical mechanisms reported in this study could be widespread across bacteria.”

16. Line 293 “Numerous inconsistent models for the fd phage capsid have been proposed over the years by X-ray fibre diffraction, ssNMR and -resolution cryo-EM”. It would be

beneficial that the authors explain what may be the source of the differences. Is it related to different constructs/reconstitutions? Please discuss.

We believe that because NMR mostly detects close-range atomic interactions, it does not perform well in obtaining the large-scale architecture of capsids, and cryo-EM at the time simply did not have sufficient resolution. It is not related to different constructs because we are using the wild-type phage (and so did the other studies). We believe that we were able to solve the structure to improved cryo-EM technology, now explicitly mentioned in the manuscript.

L363: “While lack of resolution in the past studies probably led to these inconsistencies, the advent of improved cryo-EM technology, including improved microscope optics, detectors and image analysis software¹⁰, allowed us to produce a 3.2 Å-resolution cryo-EM structure of the fd capsid.”

17. Line 328 “Pf4 droplets almost fully encapsulate rod-shaped *P. aeruginosa* and *E. coli* cells, compared to the artificial case of fd droplets, where only lateral interactions were seen.” Why is the later defined artificial? Please clarify.

We initially used the term ‘artificial’ since fd droplets are not known to occur in *Pseudomonas* biofilms and hence do not associate with cells within biofilms, unlike Pf4. We realise that terming this interaction ‘artificial’ could be confusing, so we have removed this word.

L400: “Pf4 droplets almost fully encapsulated rod-shaped *P. aeruginosa* and *E. coli* cells, compared to fd droplets, which did not fully encapsulate cells, but nevertheless closely associated with bacteria (Figure 5).”

18. Line 336 “One of the most striking results in this study was that droplet association with cells is sufficient for antibiotic protection (Figure 6), rather than complete encapsulation.” It would be useful to visualize this at higher resolution (tomography?) to be able to make this conclusion. It is possible that a small amount of phages are sufficient to block diffusion of the antibiotic.

Phages not forming liquid crystalline droplets do not confer antibiotic protection, as seen in the controls of our antibiotic protection assay, where an absence of biopolymer (which induces droplet formation) prevents protection. We also show in our revision experiments that association with an fd liquid crystalline droplet is required for reducing antibiotic diffusion into

the cell (see our response to previous comments above). It is possible that small amounts of phages surround the rest of the cell but this would be difficult to visualise as such samples are too thick to directly visualise using electron cryotomography, and even if visualised it would be difficult to draw conclusions on whether this influences antibiotic tolerance.

19. Figure 3: the standard error of the mean (SEM) plotted in transparent colour is impossible to see, so not clear if it is too small or not included in the plot.

The SEM is indeed fairly small – we now show it in grey to make it more apparent (Figure 3 and Figure S4).

“**[Figure 3] b)** Quantification of ion number at different positions in a cross section of the phage, starting from the centre of the capsid for fd (left) and Pf4 (right). Fd protein atoms are coloured blue, Pf4 coloured salmon, chloride ions red and sodium ions yellow. Protein atoms are shown for clarity. The recruitment of chloride ions to the interior of the capsid can be seen as a peak at approximately 10 Å (indicated by arrows), with a higher peak observed for fd as compared to Pf4. Mean of four repeats (simulations) is plotted for each system in bold colour, with the standard error of the mean (SEM) plotted in grey. See Figure S4e for additional analyses.”

20. Figure S1: I assume arrows in (a) is to indicate individual phages/capsids. Should be clarified in the legend. The variable local resolution in (e) is interpreted in line 711 “Slight differences in local resolution between individual capsid proteins indicate minor structural flexibility within the capsid.”, but remain unclear to me how the individual subunits, rather than different regions along each subunit could exhibit different local resolution if the map is generated with symmetrisation.

Indeed, the arrows indicate individual phages, clarified now in the legend. Minor variations in local resolution in this and other helical structures is probably caused by the helical symmetry being applied during alignment (where the orientations of each particle are measured), but not during 3D reconstruction of the final structure. This is a standard procedure implemented in RELION software (and also in other software for helical reconstruction), and as such we do not think this is significant, evidenced by the tight range of the local resolution variation within 0.4 Å. We have added a sentence to prevent any misunderstanding.

“Figure S1a) Cryo-EM image of native fd phage specimen used for structure determination. Individual phage filaments are marked by an arrow.”

“Figure S1e) Very small local resolution differences were observed in the map, indicating that most of the capsid protein is rigidly positioned in the phage.”

Reviewer #2 (Remarks to the Author):

In the Ms. entitled "Biophysical basis of phage liquid crystalline droplet-mediated antibiotic tolerance in pathogenic bacteria" the Authors claim to describe biophysical and biochemical parameters that can explain the liquid crystalline droplet mediated antibiotic mechanisms of two inovirus, namely: fd and Pf4. The Authors present a concise summary about what is the actual state of such a research avenue. There are several key points of this investigation, however, many of them are not well interconnected to each other. For example, when the Authors claim about biophysical basis, is this related to any biophysical observable in particular? In the current Ms., I cannot distinguish if such a "biophysical basis" is driven by electrostatics, mechanics, viscosity or any others. Any important outcomes and conclusions in this respect should be highlighted.

We thank the reviewer for their comments, which have helped significantly in improving the clarity of messaging of the manuscript. We realise that the 'biophysical' nature we are referring to may not have been clear in the original manuscript. We have made significant edits to the text and included new results and analyses in the manuscript to clarify what we mean by the 'biophysical basis' referred to in the title, as described in detail below. Briefly, in this manuscript we connect phage properties and interactions to the emergent material properties of liquid crystalline droplets. We find that these emergent droplet material properties influence droplet morphology and bacterial encapsulation; we find that droplet association with bacteria is required for antibiotic protection, and this suggests a role for such emergent material properties in antibiotic tolerance. Our study therefore identifies the biophysical basis for droplet-mediated antibiotic tolerance, as described in the title. The physical mechanism underlying the formation of liquid crystalline droplets is depletion interaction, an attractive force that results in orientational alignment of rods (phages in this study) in the presence of a high molecular weight polymer. We have clarified this in the revised manuscript:

Summary, L42: "We show that fd and Pf4 form liquid crystalline droplets with different morphologies, governed by emergent droplet material properties, which are determined by phage geometry and packing density. Finally, we show that droplets formed by either phage protect rod-shaped bacteria from antibiotic treatment, and that direct association with a droplet is required for protection, suggesting formation of a diffusion barrier by the droplet."

L86: "Assembly of phages into droplets is driven by a depletion interaction, an effective attractive force between rigid constituents arising from the preferential exclusion of solvent elements (depletants) from the vicinity of the larger phage constituents^{3,5}. In the presence of

high-molecular weight crowding polymers as a depletant and counterions to compensate surface charges, phages spontaneously assemble into liquid-crystalline droplets. As expected from a depletion interaction, the size of the crowding polymer positively correlates with the extent of liquid crystalline droplet formation¹. This process is thermodynamically favourable as phage alignment allows for higher degrees of freedom of the crowding polymer, increasing the total entropy^{3,6}.”

L224: “While it has previously been established that phage liquid crystalline droplets assemble due to depletion attraction interactions¹, it is unclear why the shape of the droplets differs between fd and Pf4. We hypothesised that the considerable difference in phage geometry and size (lengths 0.9 μm for fd vs 3.8 μm for Pf4) is the cause of differences in droplet morphology. To understand which phage properties governed the observed droplet morphologies, we developed a physical model of liquid-crystalline droplets (also called tactoids) containing hard rods, to link the geometrical properties of the phages to those of the droplets. First, we performed a theoretical scaling calculation¹¹, involving minimizing the free energy of a liquid crystalline droplet that accounts for surface (interfacial) and volumetric (elastic) contributions (see theory section in Methods for details).”

We further clarify throughout the manuscript that we believe liquid crystalline droplet formation is dominated by phage and droplet material properties, and that association and encapsulation may be determined by differing surface tension of the droplets:

L249: “To explain the observed difference between the Pf4 and fd curves in Figure 4g, we performed a second scaling calculation to link the material droplet properties in the prefactor of Eq. (1) to the phage geometry and packing within the droplet (see Methods). By approximating the elastic constant K and the surface tension γ (see Eq. (7), Methods), we derived the following scaling relationship:

$$\frac{R}{r} \propto \left(\rho \frac{b^2}{a} \right)^{3/5} V^{-1/5}, \quad (2)$$

where ρ is the packing fraction of the phages in the droplet ($\rho_{fd} = 0.9$ and $\rho_{Pf4} = 0.25$, as determined from analysis of tomograms; see Methods), b is the length (major axis) of the phage ($b_{fd} = 0.9 \mu\text{m}$ and $b_{Pf4} = 3.8 \mu\text{m}$), and a is the width (minor axis) of the phage ($a_{fd} = 64 \text{ \AA}$ and $a_{Pf4} = 62 \text{ \AA}$). Surprisingly, given the highly coarse-grained nature of the model, Eq. (2) predicts $C_{Pf4}/C_{fd} \sim 2.6$, a value which is within a factor of two from the fits to the experimental data (Figure 4g). We conclude that Eq. (2) contains the key determinants of tactoid shape: the shape and packing of the phages, and the overall size of the droplet. The small quantitative discrepancy between the predicted and measured values of C_{Pf4}/C_{fd} could be caused by

differences in biochemical phage-phage interactions (e.g., electrostatic surface interactions; see Figure 2) or through distinct phage geometry causing differing depletion interactions. While these interactions already feed into Eq. (2) through the phage packing fraction ρ , such interactions also affect the material properties K and γ in Eq. (1), and would therefore enter Eq. (2) as prefactors. However, the reasonably good agreement between experiment and theory suggests that differences in K and γ between Pf4 and fd tactoids are mostly determined by phage aspect ratio. Taken together, our experimental measurements and scaling theory suggest that overall droplet morphology is governed by biophysical effects through droplet size and material properties, which are determined by phage geometry and packing.”

L308: “We wondered whether the observed qualitative differences in encapsulation of bacterial cells – that Pf4 droplets more readily encapsulated the bacteria as compared to fd droplets (Figure S8f) – could be explained by biophysical phage and droplet properties within our coarse-grained framework. Based on our approximations of the bulk elastic constant K and surface tension γ in terms of the phage packing fraction ρ and phage shape (width a and length b), we predict that Pf4 droplets have a ~ 1.2 times larger K but ~ 4 times smaller surface tension γ than fd droplets. By considering the bacteria as the wetting surface for the phages, the Young-Laplace equation for the interior contact angle θ (ref⁸) gives

$$\cos(\theta) = \frac{\gamma_{b-d} - \gamma_{b-a}}{\gamma}, \quad (3)$$

where γ_{b-d} and γ_{b-a} are the respective bacteria-droplet (assumed constant) and bacteria-alginate (assumed constant) surface tensions, and γ is the alginate-droplet surface tension already defined above. Eq. (3) suggests that a higher alginate-droplet surface tension γ results in an increase in contact angle between the droplet and bacteria interfaces, i.e. reduced wetting. Therefore, we predict reduced wetting (and therefore reduced encapsulation) of bacteria by fd droplets because of their higher surface tension than Pf4 droplets. This is in line with experiments (Figure 5), further confirming that material properties, which are determined by phage shape and packing, drive the observed differences in behaviour between Pf4 and fd droplets.”

Moreover, what is the role of Inoviruses genome in the morphological changes between fd and Pf4? In particular when fd and Pf4 are compared?

In our previous publication⁴, we prepared Pf4 phages lacking a DNA genome. These genome-lacking ‘ghost’ phages were equally competent to make droplets and confer antibiotic protection to bacteria. Hence, we do not believe that the DNA itself directly influences liquid crystalline droplet formation or antibiotic protection, but rather that phage genome size controls

the length of the phage, which we suggest in this manuscript influences liquid crystalline droplet morphology. We have now clarified further the arrangement and role of the phage genome in shaping the Pf4 and fd phages, as requested by the reviewer.

L170: “Two arrangements of genomic DNA in inoviruses have previously been proposed. Either the DNA could be linear single-stranded, forming a long helical arrangement, or circular single-stranded DNA, forming a double helix-like arrangement. Initially, all inoviruses were thought to contain circular single-stranded DNA, however, in the case of Pf4, a linear single-stranded-DNA was resolved⁴. In agreement with previous work, our structural model for fd suggests a circular single-stranded arrangement, as the higher positive charge-density of the capsid protein (four basic residues in fd versus two in Pf4) suggests that twice as many negative charges of the DNA phosphate backbone are being stabilised. The length of inoviruses is proportional to their genome length, and the size of the Pf4 genome is about twice as large as the fd genome (Pf4: 12.4 kb, fd: 6.4 kb¹²), whilst their phage filament length differs by about a factor of four (0.9 μm fd¹³ vs. 3.8 μm Pf4⁴). This mismatch suggests that the genome arrangement of fd is different to Pf4, compacted by a factor of two, supporting a circular arrangement of its DNA.”

I recommend addressing the already described major points and the listed ones below (1-9) before further rounds for publication:

1. Equations 1 and 2 require a schematic representation of the very coarse theoretical model for the phages droplets.

We have produced a schematic representation and included it in Figure 4.

“**[Figure 4] f)** Schematic of the coarse-grained model whereby phages are modelled as hard rods and the phage droplets are modelled as liquid crystal droplets with bulk (elastic) and surface energetic contributions (see Eq. (2-3)). **g)** Liquid crystalline droplet aspect ratio as a function of droplet volume. Droplets, as visualised via light microscopy, follow the scaling law $R/r \propto V^{-1/5}$ as shown through lines of best fit to $R/r = CV^{-1/5}$, where C is the fitting parameter ($C_{Pf4} = 5.88 \pm 0.11 \mu m^{3/5}$ and $C_{fd} = 3.72 \pm 0.09 \mu m^{3/5}$).”

2. Section: Atomic structure line 121...: What is the Net charge of the protein for fd and Pf4?

We have now added additional information on electric charges to the revised manuscript as requested.

L164: “As a whole, the fd major coat protein is more positively charged than that of Pf4 (compare the isoelectric point of fd 6.28 versus Pf4 4.68), which is due to an increased positive charge facing the inner lumen of the phage with four basic residues present at the C-terminus compared to two in the case of Pf4 (Figure 2d).”

What about the surface charges, outside and inside the proteinaceous capsids? Which is the expected charge of the genome? What is the role of the outer surface charge and of the genome in the morphological changes of the filaments arrangements?

An electrostatic surface view is shown in Figure 2. The exact surface charge is hard to quantify and/or compare since many of the negatively charged residues in fd were found to be disordered, also observed in previous studies^{13,14}. These residues were modelled in for the calculation of the surface charge representation in Figure 2, to enable a comparison with Pf4. We added additional information on the charges within the lumen and implications for interactions of these luminal residues with DNA (see answer to previous comment). Unfortunately, the density for the genomic DNA did not support atomic model building, now discussed further in the manuscript to prevent any misunderstanding. Please also see our answer to your question about the genome above, where we discuss DNA-free viruses reported in our previous paper.

L378: “While density for DNA could be detected in our map (Figure S1f), it did not allow for unambiguous atomic model building, consistent with previous studies on the Ike phage⁷ (Figure S2).”

3. Section: Droplet encapsulation, line 254..: What is the reasoning behind the phrase: “suggesting that bacterial cell association is a biophysical process rather than being driven by specific biochemical interactions, since the surfaces of the phages as well as the outer surfaces of *E. coli* and *P. aeruginosa* are biochemically distinct.” Chemically distinct atoms could be attractive exactly if they are different, or? What type of biophysical process is the association of Bacterial cells and the Ic droplets? The proposed theory is only macroscopical (highly hypothetical) and hence of course not considering any chemical information. This point need to be clarified

We agree that this needed clarification. Regarding the interactions of the liquid crystalline droplets with the cells, we now elaborate on our rationale for using these bacteria, highlighting that they have different surfaces: *P. aeruginosa* is covered by O-antigen, an abundant cell surface polysaccharide, while *E. coli* is not. Furthermore, both bacteria have different surface proteins. Both of these cells could associate with liquid crystalline droplets, suggesting it is not a specific chemical interaction with the cell surface.

L276: “Since generic biophysical effects (shape and packing) appear to govern overall droplet morphology, we next asked how these morphologically distinct droplets interact with rod-shaped *E. coli* and *P. aeruginosa* bacteria. Previous studies have shown that liquid crystalline droplets formed by Pf4 can fully encapsulate *P. aeruginosa* cells, which correlates with protection from antibiotic treatment⁴. However, it is unclear whether this encapsulation is driven by specific chemical interactions between the phage and the cell surface, or whether it may be driven by physical factors. To answer this question, we mixed fd and Pf4 droplets with *E. coli* or *P. aeruginosa* cells, which are bacteria with significantly different surface chemistry due to different surface proteomes and the lack of a lipopolysaccharide O-antigen in *E. coli* K12¹⁵.”

In the revised manuscript, we provide additional context to the introduction on depletion-attraction, as well as in the results section and the discussion. This was also raised by reviewer 1. We used two phages with different lengths, and biochemically distinct capsid proteins to probe this system.

L81: “A unique property of inoviral phages, including fd and Pf4, is that they can spontaneously and reversibly self-assemble to form ordered but dynamic liquid crystalline droplets under crowding conditions¹⁻³. Within droplets, laterally associated phages are orientationally aligned, but not regularly ordered as in a crystal². Phages within the liquid crystalline droplets are mobile, as shown by fluorescence recovery after photobleaching (FRAP) experiments on Pf4

droplets⁴. Assembly of phages into droplets is driven by a depletion interaction, an effective attractive force between rigid constituents arising from the preferential exclusion of solvent elements (depletants) from the vicinity of the larger phage constituents^{3,5}. In the presence of high-molecular weight crowding polymers as a depletant and counterions to compensate surface charges, phages spontaneously assemble into liquid-crystalline droplets. As expected from a depletion interaction, the size of the crowding polymer positively correlates with the extent of liquid crystalline droplet formation¹.”

4. Regarding the experimental setup for using E.coli and P.aeruginosa, which are the details in terms of salt concentrations in each system mM?

The salt concentrations are equal in both cases (137 mM), now stated in the Methods section:

L624: “*P. aeruginosa* PAO1 Δ PA0728 or *E. coli* XL1, a K12 derivative, were grown to an OD₆₀₀ of 0.5 and incubated with A488-labelled phage (final concentration 1 mg/ml) and sodium alginate (final concentration 4 mg/ml) in PBS (137 mM NaCl) for 3 hours.”

how are the apparent higher charges in fd affecting the adherence of the E.coli and P.aeruginosa. Which are the main differences of both systems (E.coli and P.aeruginosa)? they need to be also depicted at least schematically in e.g. Figure 7.

In agreement with previous literature, depletion attraction occurs irrespective of charges as long as sufficient counter-ions are present in the solution^{1,5}. We chose *P. aeruginosa* PAO1 and *E. coli* K12 on one hand as they are the native host organisms for the respective phages (and hence co-evolved), and on the other hand because they have distinct chemistry on their surfaces – *P. aeruginosa* PAO1 displays O-antigen (polysaccharides) on its surface, while *E. coli* K12 does not. As suggested, we have illustrated this in the schematic Figure 7.

L276: “Since generic biophysical effects (shape and packing) appear to govern overall droplet morphology, we next asked how these morphologically distinct droplets interact with rod-shaped *E. coli* and *P. aeruginosa* bacteria. Previous studies have shown that liquid crystalline droplets formed by Pf4 can fully encapsulate *P. aeruginosa* cells, which correlates with protection from antibiotic treatment⁴. However, it is unclear whether this encapsulation is driven by specific chemical interactions between the phage and the cell surface, or whether it may be driven by physical factors. To answer this question, we mixed fd and Pf4 droplets with *E. coli* or *P. aeruginosa* cells, which are bacteria with significantly different surface chemistry

due to different surface proteomes and the lack of a lipopolysaccharide O-antigen in *E. coli* K12¹⁵.”

“Figure 7: Schematic depicting biophysical nature of phage liquid crystalline droplet-mediated antibiotic tolerance of bacteria.

The inoviruses, fd and Pf4, form liquid crystalline droplets in the presence of biopolymer with distinct morphologies dictated by phage size. Both fd and Pf4 phage liquid crystalline droplets associate with *P. aeruginosa* and *E. coli* K12, which show different cell surface chemistries. The material properties of the liquid crystalline droplets govern the association with bacteria independent of their cell surface properties. Phage liquid crystalline droplet association with bacteria form a diffusion barrier that results in increased antibiotic tolerance, which correlates with the level of bacterial cell encapsulation by the droplet.”

5. The authors need to perform contact angle measurements of the crystalline droplets, to support their analysis on the droplet formation.

We agree with the reviewer that the wetting behaviour of the droplets is an interesting aspect of this system, but we were not able to obtain accurate measurements of contact angles with the cover slip using super-resolution confocal fluorescence microscopy due to weak signal from the edge of the droplet near the glass. Moreover, we feel that wetting of bacteria by the

liquid crystalline droplets is the key property of interest, rather than wetting of the droplet on glass. However, stimulated by the reviewer's suggestion, we now infer the wetting behaviour from our scaling theory and also from our measurements of bacterial encapsulation. We have added a paragraph to discuss in detail the relationship between the scaling theory, wetting, and bacterial encapsulation, in which we use the Young Laplace equation to explain the observed differences in encapsulation (see line 283 onwards). We thank the reviewer for raising this interesting point.

L308: "We wondered whether the observed qualitative differences in encapsulation of bacterial cells – that Pf4 droplets more readily encapsulated the bacteria as compared to fd droplets (Figure S8f) – could be explained by biophysical phage and droplet properties within our coarse-grained framework. Based on our approximations of the bulk elastic constant K and surface tension γ in terms of the phage packing fraction ρ and phage shape (width a and length b), we predict that Pf4 droplets have a ~ 1.2 times larger K but ~ 4 times smaller surface tension γ than fd droplets. By considering the bacteria as the wetting surface for the phages, the Young-Laplace equation for the interior contact angle θ (ref⁸) gives

$$\cos(\theta) = \frac{\gamma_{b-d} - \gamma_{b-a}}{\gamma}, \quad (3)$$

where γ_{b-d} and γ_{b-a} are the respective bacteria-droplet (assumed constant) and bacteria-alginate (assumed constant) surface tensions, and γ is the alginate-droplet surface tension already defined above. Eq. (3) suggests that a higher alginate-droplet surface tension γ results in an increase in contact angle between the droplet and bacteria interfaces, i.e. reduced wetting. Therefore, we predict reduced wetting (and therefore reduced encapsulation) of bacteria by fd droplets because of their higher surface tension than Pf4 droplets. This is in line with experiments (Figure 5), further confirming that material properties, which are determined by phage shape and packing, drive the observed differences in behaviour between Pf4 and fd droplets.

6. I strongly recommend the authors to perform a genome structural analysis to verify the linear and circular ssDNA assumptions, this can be rapidly carried out by using a similar method as the one described here (<https://doi.org/10.3390/v13081555>); where the effects of the charges (lines 305-317) can be theoretically confirmed.

While the suggested program is for determination of secondary structure in RNA viruses, we do not expect such secondary structure formation in this case. We, however, agree with the reviewer's point that the background of this - as well as how we come to our conclusions - should be better explained in the manuscript. In the revised manuscript, we hence elaborate

on why we believe that fd has a circular single-stranded DNA in more detail. We have also explicitly state that the cryo-EM density for the DNA did not support direct atomic model building.

L170: “Two arrangements of genomic DNA in inoviruses have previously been proposed. Either the DNA could be linear single-stranded, forming a long helical arrangement, or circular single-stranded DNA, forming a double helix-like arrangement. Initially, all inoviruses were thought to contain circular single-stranded DNA, however, in the case of Pf4, a linear single-stranded-DNA was resolved⁴. In agreement with previous work, our structural model for fd suggests a circular single-stranded arrangement, as the higher positive charge-density of the capsid protein (four basic residues in fd versus two in Pf4) suggests that twice as many negative charges of the DNA phosphate backbone are being stabilised. The length of inoviruses is proportional to their genome length, and the size of the Pf4 genome is about twice as large as the fd genome (Pf4: 12.4 kb, fd: 6.4 kb¹²), whilst their phage filament length differs by about a factor of four (0.9 μm fd¹³ vs. 3.8 μm Pf4⁴). This mismatch suggests that the genome arrangement of fd is different to Pf4, compacted by a factor of two, supporting a circular arrangement of its DNA.”

7. Another important missing point in each of the results sections are the limitations of the analysis, for example the very coarse theoretical model, or if any symmetry is imposed in the cryoEM structures, among many others related to the droplet and experimental observations.

We now discuss these limitations in the revised manuscript, along with future work that should address these limitations and extend our work. Thank you for highlighting this, we agree that it is best to inform the readers about the drawbacks. We add some examples below.

L126: “Using helical reconstruction and applying C5 symmetry, a 3.2 Å-resolution map of the fd phage capsid was obtained, which allowed derivation of an atomic model of the capsid (Figures 1a-c and S1b-e, Movie S1 and Table S1).”

L262: “The small quantitative discrepancy between the predicted and measured values of $C_{\text{Pf4}}/C_{\text{fd}}$ could be caused by differences in biochemical phage-phage interactions (e.g., electrostatic surface interactions; see Figure 2) or through distinct phage geometry causing differing depletion interactions.”

L406: “Future studies could employ genomically engineered phages with various lengths to further probe the role of phage material properties on liquid crystalline droplet formation and

encapsulation. Furthermore, mutation of the surface charges could enlighten how far significant alteration of the phage surface would perturb the biophysical effects described in this study.”

8. Conclusions and further research questions are missing.

Thank you – we have now added such a section to the discussion.

L423: “Since bacteria often proliferate in environments rich in rod-like or filamentous molecules accompanied by biopolymers, for example in the biofilm matrix^{1,4,16} or on tissues covered with mucus¹⁷, our data suggest a paradigm where the geometric properties of filamentous components govern assembly into higher-order structures with corresponding material properties that shield cells from antimicrobials, imparting tolerance (Figure 7). Previous studies have suggested that entire biofilms could be considered to be a nematic, liquid-crystalline system, including alignment of cells with a high three-dimensional order^{18,19}. Our research on a minimal system consisting of cells and droplets shows how filamentous molecules assemble into liquid crystalline structures that can mediate the emergent biofilm property of antibiotic tolerance. Future research including more biochemically reconstituted components of the biofilm extracellular matrix will be needed to unambiguously prove this proposal and to delineate the exact contribution of each component in biofilm formation. Since filamentous molecules are abundant in most bacterial biofilms⁹, the biophysical mechanisms reported in this study could be widespread across bacteria.”

9. Table S1: data collection and processing statistics for fd capsid structure
—> data collection and processing statistics for Pf4 missing!!!

The structure of Pf4 was already solved as part of a previous study⁴ – in this manuscript we provide the structure of fd and thus enable a systematic comparison between these two model phages. As mentioned in the outlook, we hope that this can be further extended to include more filaments and cell types (of different morphologies) in the future.

References

- 1 Secor, P. R. *et al.* Filamentous bacteriophage promote biofilm assembly and function. *Cell host & microbe* **18**, 549-559 (2015).
- 2 Dogic, Z. & Fraden, S. Ordered phases of filamentous viruses. *Current opinion in colloid & interface science* **11**, 47-55 (2006).
- 3 Lekkerkerker, H. N. & Tuinier, R. in *Colloids and the depletion interaction* 57-108 (Springer, 2011).
- 4 Tarafder, A. K. *et al.* Phage liquid crystalline droplets form occlusive sheaths that encapsulate and protect infectious rod-shaped bacteria. *Proceedings of the National Academy of Sciences* **117**, 4724-4731 (2020).
- 5 Asakura, S. & Oosawa, F. Interaction between particles suspended in solutions of macromolecules. *Journal of polymer science* **33**, 183-192 (1958).
- 6 Petukhov, A. V., Tuinier, R. & Vroege, G. J. Entropic patchiness: Effects of colloid shape and depletion. *Current opinion in colloid & interface science* **30**, 54-61 (2017).
- 7 Xu, J., Dayan, N., Goldbourt, A. & Xiang, Y. Cryo-electron microscopy structure of the filamentous bacteriophage IKe. *Proceedings of the National Academy of Sciences* **116**, 5493-5498 (2019).
- 8 Israelachvili, J. N. *Intermolecular and surface forces*. (Academic press, 2011).
- 9 Hobley, L., Harkins, C., MacPhee, C. E. & Stanley-Wall, N. R. Giving structure to the biofilm matrix: an overview of individual strategies and emerging common themes. *FEMS microbiology reviews* **39**, 649-669 (2015).
- 10 Kühlbrandt, W. The resolution revolution. *Science* **343**, 1443-1444 (2014).
- 11 Prinsen, P. & Van Der Schoot, P. Shape and director-field transformation of tactoids. *Physical Review E* **68**, 021701 (2003).
- 12 Mai-Prochnow, A. *et al.* Big things in small packages: the genetics of filamentous phage and effects on fitness of their host. *FEMS microbiology reviews* **39**, 465-487 (2015).
- 13 Wang, Y. A. *et al.* The structure of a filamentous bacteriophage. *Journal of molecular biology* **361**, 209-215 (2006).
- 14 Colnago, L., Valentine, K. & Opella, S. Dynamics of fd coat protein in the bacteriophage. *Biochemistry* **26**, 847-854 (1987).
- 15 Liu, D. & Reeves, P. R. Escherichia coli K12 regains its O antigen. *Microbiology* **140**, 49-57 (1994).
- 16 Böhning, J. *et al.* Donor-strand exchange drives assembly of the TasA scaffold in Bacillus subtilis biofilms. *Nature Communications* **13**, 1-12 (2022).
- 17 Wagner, C., Wheeler, K. & Ribbeck, K. Mucins and their role in shaping the functions of mucus barriers. *Annu. Rev. Cell Dev. Biol* **34**, 189-215 (2018).
- 18 Hartmann, R. *et al.* Emergence of three-dimensional order and structure in growing biofilms. *Nature physics* **15**, 251-256 (2019).
- 19 Zhou, S., Sokolov, A., Lavrentovich, O. D. & Aranson, I. S. Living liquid crystals. *Proceedings of the National Academy of Sciences* **111**, 1265-1270, doi:doi:10.1073/pnas.1321926111 (2014).

REVIEWER COMMENTS

Reviewer #1 (Remarks to the Author):

The revised version of the manuscript is much improved: the authors have made great effort and succeeded in making the work more clear by expanding their introduction, clear reference to prior work, and explanation of a number of the basic concepts that form the foundation for this study.

Importantly, the authors provide experiments showing reduced antibiotic diffusion in the presence of phage meso-scale assemblies, which now stands as a plausible mechanism for the mode of antibiotic protection in this minimal system. I find the revised manuscript to be suitable for publication, pending minor corrections/clarifications in the text:

1. The now detailed explanation on the meso-scale assembly of the phage in line with colloid attraction-depletion mechanisms provides a better understanding of the work. The calculations of the effect of surface tension on encapsulation are very interesting too. However, the definition of a liquid-crystalline phase needs better clarification. Liquid-crystals imply that the constituents have some degree of rotational freedom, but do not necessarily allow for free exchange of the constituents with the environment. A liquid phase allows for the later, but the degree of exchange will depend on the visco-elastic properties of the system (in the field of biomolecular condensates, this is typically probed by fusion experiments or FRAP to support dynamic molecular exchange).

The authors now provide FRAP experiments on both phage tactoids, but these show low recovery for Pf4 (not really discussed in the results), indicating the tactoids may not be completely liquid-like. I therefore recommend that the authors rephrase their observation of "liquid-like" or "droplets", and maybe more carefully describe those assemblies as tactoids with liquid-crystalline order that exhibit different degrees of molecular exchange.

2. Along the lines of the material properties discussed above, it is interesting to see that the new FRAP experiments show that Fd tactoids have much higher recovery fraction (i.e. lower immobile fraction) than Pf4, which recovers only roughly 50% of the intensity. The $t_{1/2}$ measure provided does not properly recapitulate this difference (Figure S7), and immobile fractions should be provided in addition. These are useful parameters that serve as indicators of droplet properties, and I advise adding these to the main text as they support the author's theoretical calculations showing different materials properties. How this links to the different packing density observed by cryo-ET (fd being more highly packed but showing higher mobile fraction) is an interesting observation and should at least be mentioned/discussed.

3. In their response to reviewer 2, the authors state "In our previous publication, we prepared Pf4 phages lacking a DNA genome. These genome lacking 'ghost' phages were equally competent to make droplets and confer antibiotic protection to bacteria. Hence, we do not believe that the DNA itself directly influences liquid crystalline droplet formation or antibiotic protection, but rather that phage genome size controls the length of the phage, which we suggest in this manuscript influences liquid crystalline droplet morphology. We have now clarified further the arrangement and role of the phage genome in shaping the Pf4 and fd phages, as requested by the reviewer". The nicely provide the background on this in the revision, but should also include a discussion of these previous findings of tactoids in the absence of the genome in relation to the current work (and whether any difference are observed).

4. The logic of the statement in the abstract: "...form liquid crystalline droplets with different morphologies, governed by emergent droplet material properties, which are determined by phage geometry and packing density." is somewhat unclear and may require rewording. I believe that what I understand is that the different morphologies are the result of different phage geometry and packing, which give rise to different emergent properties.

Reviewer #2 (Remarks to the Author):

This new version of the Ms. contains several important points addressed after my first revision round. However, at this stage I cannot accept the current version before addressing in detail the following points.

2. What is the Net charge of the protein for fd and Pf4? What about the surface charges, outside and inside the proteinaceous capsids? Which is the expected charge of the genome? What is the role of the outer surface charge and of the genome in the morphological changes of the filaments arrangements?

Note: The answer " Unfortunately, the density for the genomic DNA did not support atomic model building, now discussed further in the manuscript to prevent any misunderstanding."

This is a very important point regarding the genome of the inovirus. In fact, I do not understand why the Authors claim in point 1. of their answer that the genome has no effect in the phage. They claim Reference 4.

They do claim the genome has no effect, although some disorder negatively charged residues are the reasons for not reproducing accurate electrostatics. Are those not related? I insist that a proper explanation of this must be given. At this stage I find it very superficial and not suitable for the journal. [Also related to Questions 6 of Reviewer 1 and not satisfactory answer given]

Moreover, point 6. [I strongly recommend the authors to perform a genome structural analysis to verify the linear and circular ssDNA assumptions, this can be rapidly carried out by using a similar method as the one described here (<https://doi.org/10.3390/v13081555>); where the effects of the charges (lines 305-317) can be theoretically confirmed.] of the review about the proper literature discussing the genome reconstruction in viruses, which was not properly read by the Authors, this is not about secondary structure, is about reconstruction of genomes also of ssDNA or ssRNA. Why? well, because many of those provide necessary packaging signals for the capsid stability. I suggest to the Authors to review the virus assembly literature again. The previously proposed literature of Reference 4 was clearly not matching any atomistic model, hence you need to open the discussion to the REAL genome reconstruction as proposed in (<https://doi.org/10.3390/v13081555>).

5. The authors need to perform contact angle measurements of the crystalline droplets, to support their analysis on the droplet formation.

Again here, the answer is focused on continuum models not providing any shape of the droplets, however, in the literature there are clear methods to this by molecular simulations, given a much superior insight. Like in this work: <https://doi.org/10.1063/5.0121076> or also this <https://doi.org/10.1021/acs.jpcc.2c01599>. Those methods need to be discussed and presented within the Ms. and not just let the readership be nebulous that this cannot be done due to resolution... simulation efforts are important in this regard, and hence should be discussed and presented.

Response to reviewers for Böhning *et al* NCOMMS-23-03978-B

Green: Response Text

Blue: Manuscript excerpt

REVIEWER COMMENTS

Reviewer #1 (Remarks to the Author):

The revised version of the manuscript is much improved: the authors have made great effort and succeeded in making the work more clear by expanding their introduction, clear reference to prior work, and explanation of a number of the basic concepts that form the foundation for this study.

Importantly, the authors provide experiments showing reduced antibiotic diffusion in the presence of phage meso-scale assemblies, which now stands as a plausible mechanism for the mode of antibiotic protection in this minimal system. I find the revised manuscript to be suitable for publication, pending minor corrections/clarifications in the text:

1. The now detailed explanation on the meso-scale assembly of the phage in line with colloid attraction-depletion mechanisms provides a better understanding of the work. The calculations of the effect of surface tension on encapsulation are very interesting too. However, the definition of a liquid-crystalline phase needs better clarification. Liquid-crystals imply that the constituents have some degree of rotational freedom, but do not necessarily allow for free exchange of the constituents with the environment. A liquid phase allows for the later, but the degree of exchange will depend on the visco-elastic properties of the system (in the field of biomolecular condensates, this is typically probed by fusion experiments or FRAP to support dynamic molecular exchange).

The authors now provide FRAP experiments on both phage tactoids, but these show low recovery for Pf4 (not really discussed in the results), indicating the tactoids may not be completely liquid-like. I therefore recommend that the authors rephrase their observation of “liquid-like” or “droplets”, and maybe more carefully describe those assemblies as tactoids with liquid-crystalline order that exhibit different degrees of molecular exchange.

As requested, we now refer to these assemblies as ‘tactoids’ instead of ‘liquid crystalline droplets’ throughout the manuscript and discuss their molecular exchange properties.

Line 222: “Despite differences in packing, phages were found to show similar mobility within both fd and Pf4 tactoids as shown by FRAP experiments (Figure S7, Movie S3). Interestingly, however, the immobile fraction of Pf4 was found to be considerably higher than for fd (13.9% for fd vs 45.6% for Pf4; Figure S7), which suggests that these tactoids, despite exhibiting local liquid crystalline order, show differing degrees of molecular exchange.”

2. Along the lines of the material properties discussed above, it is interesting to see that the new FRAP experiments show that Fd tactoids have much higher recovery fraction (i.e. lower immobile fraction) than Pf4, which recovers only roughly 50% of the intensity. The t1/2 measure provided does not properly recapitulate this difference (Figure S7), and immobile fractions should be provided in addition. These are useful parameters that serve as indicators of droplet properties, and I advise adding these to the main text as they support the author's theoretical calculations showing different materials properties. How this links to the different packing density observed by cryo-ET (fd being more highly packed but showing higher mobile fraction) is an interesting observation and should at least be mentioned/discussed.

We have measured the immobile fractions and provide this data in Figure S7h. We now discuss in the main text that Pf4 is less mobile than fd and that this may reflect different materials properties.

Line 224: “Interestingly, however, the immobile fraction of Pf4 was found to be considerably higher than for fd (13.9% for fd vs 45.6% for Pf4; Figure S7), which suggests that these tactoids, despite exhibiting local liquid crystalline order, show differing degrees of molecular exchange.”

Line 406: “Interestingly, FRAP experiments show a higher immobile fraction of Pf4 compared to fd. This is despite tighter packing of phages in fd tactoids, together underlining the differing material properties between fd and Pf4 tactoids.”

3. In their response to reviewer 2, the authors state “In our previous publication, we prepared Pf4 phages lacking a DNA genome. These genome lacking ‘ghost’ phages were equally competent to make droplets and confer antibiotic protection to bacteria. Hence, we do not believe that the DNA itself directly influences liquid crystalline droplet formation or antibiotic protection, but rather that phage genome size controls the length of the phage, which we suggest in this manuscript influences liquid crystalline droplet morphology. We have now clarified further the arrangement and role of the phage genome in shaping the Pf4 and fd phages, as requested by the reviewer”. The nicely provide the background on this in the revision, but should also include a discussion of these previous findings of tactoids in the absence of the genome in relation to the current work (and whether any difference are observed).

Thank you for the suggestion – we now mention this in the main text. We believe this addition will also go towards addressing Reviewer #2's question regarding the role of phage DNA.

Line 103: “Pf4 was subsequently found to form tactoids in the extracellular matrix of biofilms, where it encapsulates bacterial cells, allowing them to tolerate antibiotic treatment^{4,15}. Atomic structure of the Pf4 phage capsid, tactoid formation properties and protection from antibiotic treatment was unaffected regardless of whether the genome was still present within the phage filament, showing that the genome did not affect the observed properties of tactoids⁴.”

4. The logic of the statement in the abstract: “...form liquid crystalline droplets with different morphologies, governed by emergent droplet material properties, which are

determined by phage geometry and packing density.” is somewhat unclear and may require rewording. I believe that what I understand is that the different morphologies are the result of different phage geometry and packing, which give rise to different emergent properties.

We have reworded this sentence in accordance with the reviewer’s suggestion:

Line 43: “We show that fd and Pf4 form tactoids with different morphologies that arise from different phage geometries and packing densities, which in turn give rise to different tactoid emergent properties.”

Reviewer #2 (Remarks to the Author):

This new version of the Ms. contains several important points addressed after my first revision round. However, at this stage I cannot accept the current version before addressing in detail the following points.

2. What is the Net charge of the protein for fd and Pf4? What about the surface charges, outside and inside the proteinaceous capsids? Which is the expected charge of the genome? What is the role of the outer surface charge and of the genome in the morphological changes of the filaments arrangements?

Note: The answer " Unfortunately, the density for the genomic DNA did not support atomic model building, now discussed further in the manuscript to prevent any misunderstanding."

This is a very important point regarding the genome of the inovirus. In fact, I do not understand why the Authors claim in point 1. of their answer that the genome has no effect in the phage. They claim Reference 4.

Sorry about this, we feel that there has been a huge misunderstanding in this round of reviews, particularly related to the phage genome. In previous versions of this manuscript, we had not claimed anywhere that the genome of the phage had no effect on the phage, which is why we are surprised that the reviewer is so intensively focused on the phage genome, which is not the central focus of this manuscript. We have tried to clarify these statements in this version, providing previous results, new results and textual updates.

They do claim the genome has no effect, although some disorder negatively charged residues are the reasons for not reproducing accurate electrostatics. Are those not related? I insist that a proper explanation of this must be given. At this stage I find it very superficial and not suitable for the journal. [Also related to Questions 6 of Reviewer 1 and not satisfactory answer given]

Apologies for the confusion, we do not intend to claim that the genome has no effect on the assembly/life cycle of the inovirus. We are referring to protection from antibiotic treatment, which we previously probed by producing Pf4 phage lacking a genome, which can be generated using lithium chloride treatment.

Experiments with these genome-less phages, previously termed 'ghost' phages, showed that ghost phages can still form liquid crystalline droplets, and they still protect *P. aeruginosa* from death during antibiotic treatment. Reviewer #1 also suggested to mention this in the main text, which we have now added.

Line 103: "Pf4 was subsequently found to form tactoids in the extracellular matrix of biofilms, where it encapsulates bacterial cells, allowing them to tolerate antibiotic treatment^{4,15}. Atomic structure of the Pf4 phage capsid, tactoid formation properties and protection from antibiotic treatment was unaffected regardless of whether the genome was still present within the phage filament, showing that the genome did not affect the observed properties of tactoids⁴."

To remove any misunderstandings, we summarise below our previously published results on Pf4 capsid structure, tactoid formation and antibiotic protection with and without the phage genome. Again, we are not claiming that the genome has no effect on the phage, but you will see that the genome has no discernible effect in the properties measured below.

Capsid structure of Pf4 phage without ssDNA is identical to Pf4 with ssDNA

Pf4 ghosts (without ssDNA present) assemble into liquid crystalline droplets

Pf4 ghosts (without ssDNA present) protect *P. aeruginosa* against Tobramycin

Figure R1: Previous work on the structure, liquid crystalline droplet formation, and antibiotic protection of lithium chloride-treated DNA-free ('ghost') Pf4 phages. Adapted from Tarafder *et al.* (A-B) Cryo-EM 2D class averages of normal, ssDNA-containing (left) and ghost, ssDNA-free Pf4 phage (right). (C-D) Structure of the ghost phage capsid is nearly identical to the normal phage capsid. (E-G) Mixing of normal Pf4 with ghost Pf4 shows that phage incorporation into droplets is independent of ssDNA. (H) Antibiotic protection assay showing that Pf4 ghost phages provide similar protection as normal Pf4 phage against tobramycin treatment.

Nevertheless, given the reviewers concern about the effect of the genome, we also provide new experimental data on Pf4 ghost phages. We have now repeated the antibiotic uptake assay (Figure 6) with Pf4 ghost phages (i.e. phage capsids with the DNA genome experimentally removed). We see that tactoids composed of Pf4 ghost phages, without the DNA genome present, can encapsulate cells and show a similar decrease in antibiotic uptake as compared to cells encapsulated with tactoids composed of Pf4 phage with the DNA genome present (Figure 6i-k)

Legend to Figure 6: “**c-j**) Fluorescence and light microscopy images of Alexa 488-labelled fd, Pf4 and Pf4 ghost phage tactoids incubated with Texas Red-labelled gentamicin (GTTR) for 4 hours. Shown are Alexa488 phage signal (Cyan – fd, Magenta – Pf4) and pseudocoloured cells (yellow) as determined through brightfield light microscopy (left), and Texas Red signal corresponding to uptake of the fluorescently labelled antibiotic GTTR by cells (right). Numbering indicates site of cells in corresponding images. **k**) Plot quantifying GTTR uptake after 4 hours respectively in conditions indicated on the x-axis. Bar shows the mean of three independent experiments and error bars represent standard deviation. Association with both Pf4 and fd tactoids results in significantly reduced antibiotic uptake as compared to a control with no phage (fd $P_{\text{value}} < 0.0001$, Pf4 $P_{\text{value}} < 0.0001$), and as compared to cells in the same sample that are not tactoid-associated (fd $P_{\text{value}} < 0.0001$, Pf4 $P_{\text{value}} < 0.0001$). No significant difference (n.s.) in antibiotic uptake was observed between Pf4 and Pf4 ghost tactoids.”

Line 353: “The same decrease in antibiotic uptake was observed when Pf4 ghost phage, chemically treated with lithium chloride to remove phage DNA was used

(Figure 6i-k), suggesting this effect is independent of the presence of the genome within the phage.”

Line 438: “This effect was independent of whether the DNA genome was still present within the phage filaments, suggesting that while the genome plays a crucial role in phage assembly and lifecycle, its absence does not alter diffusion of the antibiotic.”

The negatively charged, disordered N-terminal residues mentioned by the reviewer are present on the phage surface and hence do not interact with the DNA genome. We apologise if this was not clear in the manuscript and have clarified the main text:

Line 135: “The helix is terminated at the N-terminus of the mature protein by a proline residue (P6), and residues 1-5 of the capsid protein, which are located on the phage surface, are disordered (Figure 1c).”

Moreover, point 6. [I strongly recommend the authors to perform a genome structural analysis to verify the linear and circular ssDNA assumptions, this can be rapidly carried out by using a similar method as the one described here (<https://doi.org/10.3390/v13081555>); where the effects of the charges (lines 305-317) can be theoretically confirmed.] of the review about the proper literature discussing the genome reconstruction in viruses, which was not properly read by the Authors, this is not about secondary structure, is about reconstruction of genomes also of ssDNA or ssRNA. Why? well, because many of those provide necessary packaging signals for the capsid stability. I suggest to the Authors to review the virus assembly literature again. The previously proposed literature of Reference 4 was clearly not matching any atomistic model, hence you need to open the discussion to the REAL genome reconstruction as proposed in (<https://doi.org/10.3390/v13081555>).

We agree that the topology of the genome is still an important question in the field to be answered. While we believe that a genome reconstruction would be interesting and may provide insights into capsid assembly, performing Monte Carlo simulations as laid out in the cited study would constitute a separate work in its own, which is beyond our expertise and capabilities. We now, however, discuss the possibility of such simulations in the manuscript, citing the corresponding work.

Line 394: “Further studies should probe the organisation of inoviral phage DNA genomes, as demonstrated previously for RNA viruses using Monte Carlo simulations⁴³, which could provide important insights into the organisation of the DNA genome and its interaction with the capsid.”

5. The authors need to perform contact angle measurements of the crystalline droplets, to support their analysis on the droplet formation.

Again here, the answer is focused on continuum models not providing any shape of the droplets, however, in the literature there are clear methods to this by molecular simulations, given a much superior insight. Like in this work: <https://doi.org/10.1063/5.0121076> or also this

<https://doi.org/10.1021/acs.jpcc.2c01599>. Those methods need to be discussed and presented within the Ms. and not just let the readership be nebulous that this cannot be done due to resolution... simulation efforts are important in this regard, and hence should be discussed and presented.

Thank you – we agree that such efforts could be insightful and now present and discuss these works in the manuscript.

Line 422: “In addition, simulation techniques could advance our understanding of phage tactoid interaction with bacterial cells. Previous studies have utilised molecular simulations and theoretical analyses to study the interactions of surfactants to various interfaces, enabling the determination of contact angles and line tensions^{44,45}. Such methods may provide further mechanistic insights into tactoid association with bacterial cells.”